# Soil greenhouse gas fluxes from tropical coastal wetlands and alternative agricultural land uses

Naima Iram[1], Emad Kavehei[1], Damien. T. Maher[2], Stuart. E. Bunn[1], Mehran Rezaei Rashti[1], Bahareh Shahrabi Farahani[1], Maria Fernanda Adame[1]

[1]Australian Rivers Institute, Griffith University, Nathan, QLD, 4111, Australia
[2]School of Environment, Science and Engineering, Southern Cross University, Lismore, NSW, 2480, Australia

*Correspondence to*; Naima Iram (naima.iram@griffithuni.edu.au)

**Abstract.** Coastal wetlands are essential for regulating the global carbon budget through soil carbon sequestration and greenhouse gas fluxes (GHG: $CO_2$, $CH_4$ and $N_2O$). The conversion of coastal wetlands to agricultural land alters these fluxes' magnitude and direction (uptake/release). However, the extent and drivers of change of GHG fluxes is still unknown for many tropical regions. We measured soil GHG fluxes from three natural coastal wetlands: mangroves, saltmarsh, and freshwater tidal forests, and two alternative agricultural land use, sugarcane farming and pastures for cattle grazing (ponded and dry conditions). We assessed variations throughout different climatic conditions (dry-cool, dry-hot and wet-hot) within two years of measurements (2018-2020) in tropical Australia. The wet pasture had by far the highest $CH_4$ emissions with $1,231 \pm 386$ mg m$^{-2}$ d$^{-1}$, which were 200-fold higher than any other site. Dry pastures and sugarcane were the highest emitters of $N_2O$ with $55 \pm 9$ mg m$^{-2}$ d$^{-1}$ (wet-hot period) and $11 \pm 3$ mg m$^{-2}$ d$^{-1}$ (hot-dry period, coinciding with fertilisation), respectively. Dry pastures were also the highest emitters of $CO_2$ with $20 \pm 1$ g m$^{-2}$ d$^{-1}$ (wet-hot period). The three coastal wetlands measured had lower emission, with saltmarsh uptake of $-0.55 \pm 0.23$ of $N_2O$ and $-1.19 \pm 0.08$ g m$^{-2}$ d$^{-1}$ of $CO_2$ during the dry-hot period. During the sampled period, sugarcane and pastures had higher total cumulative soil GHG emissions ($CH_4 + N_2O$) of 7,142 and 56,124 CO$_{2eq}$ kg ha$^{-1}$ y$^{-1}$ compared to coastal wetlands with 144 to 884 CO$_{2\text{-eq}}$ kg ha$^{-1}$ y$^{-1}$. Restoring unproductive sugarcane land or pastures (especially ponded ones) to coastal wetlands could provide significant GHG mitigation.

## 1 Introduction

Coastal wetlands are found at the interface of terrestrial and marine ecosystems and account for 10% of the global wetland area (Lehner and Döll, 2004). They are highly productive and provide various ecosystem services such as water quality improvement, biodiversity, and carbon sequestration (Duarte et al., 2013). For instance, mangroves can accumulate five times more soil carbon than terrestrial forests (Kauffman et al., 2020). However, the high productivity and anoxic soil conditions that promote carbon sequestration can also favour potent greenhouse gas emissions (GHGs), including $CO_2$, $CH_4$ and $N_2O$ (Whalen, 2005; Conrad, 2009).

The GHG emissions in coastal wetlands primarily result from microbial processes in the soil-water-atmosphere interface ((Bauza et al., 2002) Whalen, 2005). The emission of $CO_2$ is a result of respiration, where fixed carbon by photosynthesis is partially released back into the atmosphere (Oertel et al., 2016). Emissions of $CH_4$ result from anaerobic and aerobic respiration by methanogenic bacteria, mostly in waterlogged conditions (Angle et al., 2017; Saunois et al., 2020). Finally, $N_2O$ emissions are caused by denitrification in anoxic conditions and nitrification in aerobic soils, both driven by nitrogen content and soil moisture (Ussiri and Lal 2013). Thus, the total GHG emissions from a wetland are driven by environmental conditions that favour these microbial processes, all of which result in highly variable emissions from wetlands worldwide (Kirschke et al., 2013; Oertel et al., 2016).

Despite potential high GHG emissions from coastal wetlands, these are likely to be lower than those from alternative agricultural land use (Knox et al., 2015), which emit GHGs from their construction throughout their productive lives. Firstly, when wetlands are converted to agricultural land, the oxidation of sequestered carbon in the organic-rich soils release significant amounts of $CO_2$ (Drexler et al., 2009; Hooijer et al., 2012). Secondly, removing tidal flow and converting coastal wetlands to freshwater systems, such as during the creation of ponded pastures, dams or agricultural ditches, can result in very high $CH_4$ emissions (Deemer et al., 2016; Grinham et al., 2018; Ollivier et al., 2019). For instance, agricultural ditches contribute up to 3% of the total anthropogenic $CH_4$ emissions globally (Peacock et al., 2021). Finally, the use of fertilisers significantly increases $N_2O$ emissions (Rezaei Rashti et al., 2015). Thus, emissions of GHG from land-use change can be mitigated through the reversal of these activities, for instance, reduction of fertiliser use and the reinstallation of tidal flow on unused agricultural land (Kroeger et al., 2017; Rezaei Rashti et al., 2015).

This study measured the annual GHG fluxes ($CO_2$, $CH_4$ and $N_2O$) from three natural coastal wetlands (mangroves, saltmarsh and freshwater tidal forests) and two agricultural land use sites (sugarcane plantation and pasture) in tropical Australia. The objectives were to assess the difference in GHG fluxes throughout different seasons that characterise tropical climates (dry-cool, dry-hot and wet-hot) and identify environmental factors associated with these GHG fluxes. These data will

inform emission factors for converting wetlands to agricultural land uses and vice versa, filling in a knowledge gap identified in Australia (Baldock et al., 2012) and tropical regions worldwide (IPCC, 2013).

## 2 Materials and Methods

### 2.1 Study sites

The study area is located within the Herbert River catchment in Queensland, northeast Australia (Fig 1a). The region has a tropical climate with a mean monthly minimum temperature from 14 to 23˚C and mean monthly maximum temperature from 25 to 33˚C (Australian Bureau of Meteorology, ABM, 2020; 1968-2020; Table S3). The average rainfall is 2,158 mm y$^{-1,}$ with
the highest values of 476 mm during February (ABM 2020; 1968-2020; Table S3). The Herbert basin covers 9,842 km$^2$, from which 56% is grazing, 31% is conserved wetlands and forestry, 8% is sugarcane, and 4% is other land uses (Department of Science and Environment, DES, Wetland*Info*, 2020). Wetlands in this region were heavily deforested in the past century (1943-1996) due to rapid agricultural development, primarily for sugarcane farming (Griggs, 2018). Before clearing, the land was mostly covered by rainforest and coastal wetlands, mainly *Melaleuca* forest, grass and sedge swamps (Johnson et al., 1999).


        We selected five sites, including three natural coastal wetlands (Fig. 1): a mangrove forest (18º 53' 42″ S, 146º 15' 51″ E), a saltmarsh (18º 53' 43″ S, 146º 15' 52″ E) and a freshwater tidal forest (18º 53' 45″ S, 146º 15' 52″ E), and two common agricultural land use types of the region, a sugarcane crop (18º 53' 44.6″ S, 146º 15' 53.2″ E) and a pasture for fodder grazing. The pasture had different levels of inundation, some areas were covered with shallow ponds (50-100 cm depth), some were
wet (hereafter "wet pasture"; '18º 43' 8″ S, 146º 15' 50″ E) and others were dry (hereafter, "dry pasture"; 18º 43' 7″ S, 146º 15' 50″ E).  The natural coastal wetlands and the sugarcane site were located within the same property < 200 m apart at Insulator Creek (Fig. 1B), while the ponded pasture was 20 km north at Mungalla Station (Fig. 1A). The mangroves were dominated by *Avicennia marina* with few plants of *Rhizophora stylosa,* and the saltmarsh was dominated by *Sueda salsa* and *Sporobolus spp*.  Landwards, the freshwater tidal freshwater forest, a wetland commonly known as "tea tree swamp", was dominated by
*Melaleuca quinquenervia* trees. While the mangroves and saltmarsh are directly submerged by tides (5- 30 cm), the tidal freshwater forest is indirectly affected by tidal fluctuations, such as during large spring tides, when tidal water can push groundwater above the forest thus forming "supra-tidal" wetlands.  The coastal wetlands were adjacent to a sugarcane farm of ~110 ha (Fig. 1b). The sugarcane is fertilised once a year with urea at a rate of 135 kg N ha$^{-1}$ and harvested during May-June, while the foliage is left on the soil surface (trash blanket) after harvest. The ponded pastures in Mungalla Station is a 250 ha
farm and support ~900 cattle by providing fodder during dry periods. The selected ponded pastures were covered by *Eichhornia crassipes* (water hyacinth) and *Hymenachnae amplexicaulis* (Fig. 1g-h). Each of the five sites was sampled during three periods: dry-cool (May-September), dry-hot (October-December) and wet-hot (January-April; Table 1). During each time, soil physicochemical properties and GHG fluxes were measured as detailed below.

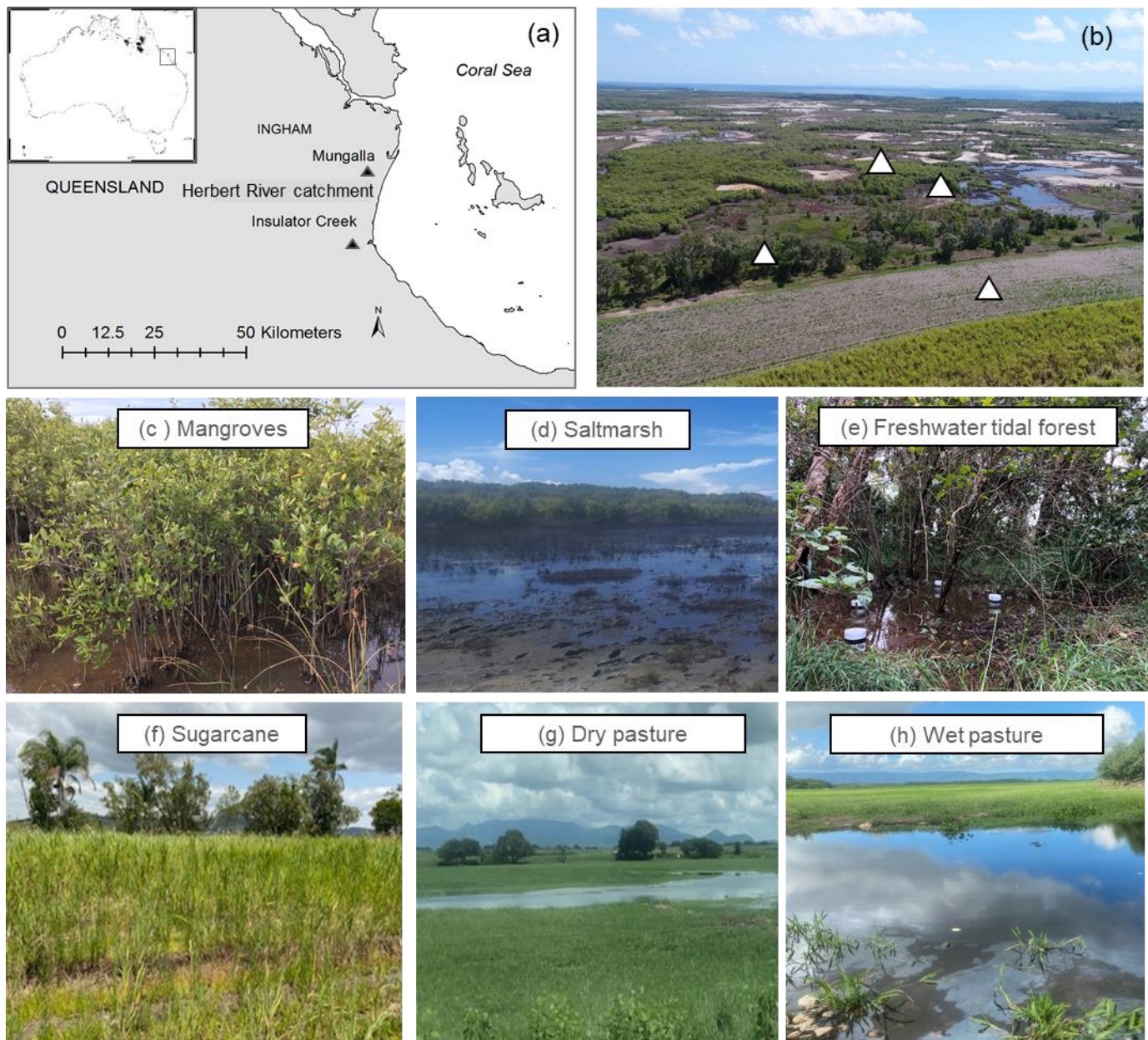

Figure 1: a) Location of sampling sites (Insulator Creek and Mungalla) in the Herbert River catchment, northeast Australia, (b) natural wetlands adjacent to sugarcane in Insulator Ck and sampling locations, and (c) mangroves, (d) saltmarsh, (e) freshwater tidal forest, (f) sugarcane, (g) dry pasture and (h) wet pasture. Pictures by N. Iram and MF Adame.



Table 1. Mean daily air temperature and rainfall range (Ingham, weather station 32078) during sampling.

| Season | Study period | Daily min temperature (°C) | Daily max temperature (°C) | Rainfall (mm d$^{-1}$) |
|---|---|---|---|---|
| Dry-cool | 17/06/2018 | 13.4 - 14.6 | 27.7 - 28.2 | 0 |
| Dry-hot | 23-29/10/2018 | 15.7 - 21.1 | 32.2 - 36.2 | 0 |
| Dry-cool | 31/05 to 6/06/2019 | 10.9 - 17.5 | 21.6 - 28.2 | 0-25 |
| Wet-hot | 17-22/02/2020 | 23.9 - 25.3 | 33.6 - 34.5 | 0-86 |

## 2.2 Soil sampling and analysis of physicochemical properties

Soil physicochemical characteristics were measured in composite soil samples next to each gas chamber location (n = 5; 0-30 cm) for all study sites during the dry-hot season. The samples were obtained by inserting an open steel corer to a depth of 30 cm; the core was divided into three depths: 0-10, 10-20 and 20-30 cm. Soil samples were oven-dried at 105 ºC for 48 h to determine volumetric water content through gravimetric analysis. The volumetric water content was divided by total soil porosity to determine water-filled pore space (WFPS). Total soil porosity was calculated using an equation from Rezaei Rashti
et al. (2015, Eq.1):

$$\text{Total soil porosity} = 1 - [(\text{soil bulk density (mg cm}^{-3})/ 2.65)]$$

Equation 1.

Soil texture analysis (% sand, silt, clay) was done with a simplified method for particle size determination (Kettler et al., 2001). Soil electrical conductivity (EC) and pH were measured using a conductivity meter (WP-84 TPS, Australia) in soil/water slurry at 1:5. Soil subsamples were air-dried, sieved (2 mm), ground (Retch™ mill) and analysed for %N and %C using an elemental analyser connected to a gas-isotope ratio mass spectrometer (EA-Delta V Advantage IRMS, Griffith University). Additionally, soil samples from the top 10 cm were collected during each sampling to measure gravimetric soil
moisture content and bulk density.

## 2.3 Greenhouse gas fluxes

We measured GHG fluxes ($CO_2$, $CH_4$ and $N_2O$) at each site for three consecutive days during each sampling period except for the dry-cool period of 2018, when mangroves, saltmarsh and sugarcane were surveyed for one day. The sampling was done
between 9:00 to 11:00 am, representing the mean daily temperatures, thus, minimising variability of cumulative seasonal fluxes

based on intermittent manual flux measurements (Reeves et al., 2016). Additionally, we assessed the variability of our measurements with tidal inundation in mangroves and saltmarsh, which were regularly inundated (~10-30 cm) during the hot-dry season. However, due to logistic constraints, further sampling was conducted only during low tides.

We used static, manual gas chambers made of high-density, round polyvinyl chloride pipe, which consisted of two units: a base (r =12 cm, h =18 cm) and a detachable collar (h =12 cm; (Hutchinson and Mosier, 1981; Kavehei et al., 2021). The chambers had lateral holes that could be left covered with rubber bungs at low water levels and left open at high water levels to allow water movement between sampling events. When the wetlands were inundated for the experiments, we used PVC extensions (h = 18 cm). Five chambers were set ~ 5cm deep in the soil, separated one to two meters from each other,

selectively located on soil with minimal vegetation, roots, and crab burrows. The chambers were set one day before sampling to minimise installation disturbance during the experiment (Rezaei Rashti et al., 2015). We were careful not to tramp around the chambers during installation and sampling. The fact that emissions were not significantly different among days ($p$ >0.05) provided us with confidence that disturbance due to installation was not problematic.

At the start of the experiment, gas chambers were closed. A sample was taken at time zero and then after one hour with a 20 ml syringe and transferred to a 12 mL-vacuumed container (Exetainer, Labco Ltd., High Wycombe, UK). During the dry-hot season, linearity tests of GHG fluxes with time were conducted by sampling at 0, 20, 40 and 60 min at all chambers (Rezaei Rashti et al., 2016). For the rest of the experiments, linearity tests were performed in one chamber per site; $R^2$ values were consistently above 0.70. During each experiment, soil temperature was measured next to each chamber. For each

experiment, the base depth was recorded from five points within each chamber to calculate headspace volume. The obtained volumetric unit concentrations were converted to mass-based units using the Ideal Gas Law (Hutchinson and Mosier, 1981).

The GHG concentrations of all samples were analysed within two weeks of sampling with a gas chromatograph (Shimadzu GC-2010 Plus). For $N_2O$ analysis, an electron capture detector was used with helium as the carrier gas, while $CH_4$

was analysed on a flame ionisation detector with nitrogen as the carrier gas. For $CO_2$ determination, the gas chromatograph was equipped with a thermal conductivity detector. Peak areas of the samples were compared against standard curves to determine concentrations (Chen et al., 2012). Seasonal cumulative GHG fluxes were calculated by modifying the equation described by Shaaban et al. (2015; Eq. 2):

$$\text{Seasonal cumulative GHG fluxes (mg or μg m} - 2 \text{ yr} - 1) = \sum_{i=1}^{n} (\text{Ri} \times 24 \times \text{Di} \times 17.38)$$

150                                                                                  Equation 2

Where;

Ri = Gas emission rate (mg m$^{-2}$ hr$^{-1}$ for $CO_2$ and μg m$^{-2}$ hr$^{-1}$ for $CH_4$ and $N_2O$),

Di = Mean daily GHG emission rate in a season (mg m$^{-2}$ d$^{-1}$ for $CO_2$ and μg m$^{-2}$ d$^{-1}$ for $CH_4$ and $N_2O$) during low tide

17.38 = number of weeks in each season, assuming these conditions were representative of the annual cycle (see Table 1).

Annual cumulative soil GHG fluxes ($CH_4$ + $N_2O$) were calculated by integrating cumulative seasonal fluxes. These estimations did not account for soil $CO_2$ values as our methodology with dark chambers only accounted for emissions from respiration and excluded uptake by primary productivity. The $CO_{2\text{-equivalent}}$ ($CO_{2\text{-eq}}$) values were estimated by multiplying $CH_4$ and $N_2O$ emissions by 25 and 298, respectively (Solomon, 2007), which represented the radiative balance of these gases (Neubauer, 2021). For annual cumulative soil GHG flux calculations from coastal wetlands, we used GHG fluxes measured during low tide; therefore, our values did not incorporate the effect of tidal fluctuations. The spatial and temporal replication of this study targeted spatial variation within soil type (< 50 cm, five chambers), days (three days per sampling) and seasons (three seasons per year). However, our replication within land use and wetland type was limited; thus, generalisations for all wetlands and land uses should be done acknowledging this limitation.

## 2.4 Statistical analyses

GHG flux data were tested for normality through Kolmogorov-Smirnov and Shapiro-Wilk tests. The data was then analysed for spatial and temporal differences with one-way Analyses of Variance (ANOVA), where site and season were the predictive factors and the replicate (chamber) was the random factor of the model. When data were not normal, they were transformed (log10 or $1/x$) to comply with the assumptions of normality and homogeneity of variances. Some variables were not normally distributed despite transformations and were analysed with the non-parametric Kruskal-Wallis test and Mann-Whitney U Test. A Pearson correlation test was run to evaluate the correlation of GHG with measured environmental factors. Analyses were done with SPSS (v25, IBM, New York, USA), and values are presented as mean ± standard error (SE).

## 3 Results

### 3.1 Soil physicochemical properties

Soil physical and chemical parameters (mean values 0-30 cm) varied among sites (Table 2, see full results of statistical analyses in Supplementary Material). As expected, gravimetric moisture content was highest in the coastal wetlands and wet pasture (> 26%) and lowest in the sugarcane and the dry pasture (< 14%). All soils were acidic, especially the freshwater tidal forest and the wet pastures with values < 5 throughout the sediment column; mangroves had the highest pH with 6.0 ± 0.1. The lowest EC was recorded in the pastures (247 ± 38 and 190 ± 39 µS cm$^{-1}$ for the dry and wet pasture, respectively), and highest in the three natural coastal wetlands with 1,418 ± 104, 8,049 ± 276 and 8,930 ± 790 µS cm$^{-1}$ for the tidal freshwater forest, saltmarsh and mangroves, respectively.

Soil bulk density was highest in sugarcane (1.5 ± 0.1 g cm$^{-3}$) and lowest in the freshwater tidal forest (0.6 ± 0.1 g cm$^{-3}$). For all sites, %C was highest in the top 10 cm of the soil and decreased with depth, with the highest values in the freshwater

tidal forest (5.1 ± 0.6%) and lowest in the saltmarsh (1.2 ± 0.1%). Soil %N ranged from 0.1 ± 0.0 to 0.4 ± 0.1% in all sites,
except in the freshwater tidal wetland, where it reached values of 0.6 ± 0.0% in the top 10 cm (Table 2).

## 3.2 Greenhouse gas fluxes

Soil emissions for $CO_2$ were significantly different among sites and times of the year ($t$ =155.09, n =237, $p < 0.001$; Fig. 2a).
The highest $CO_2$ emissions were measured during the wet-hot period in the dry pasture, where values reached 20308 ± 1951
mg m$^{-2}$ d$^{-1}$ while the lowest values were measured in the saltmarsh, the only site that acted as a sink of $CO_2$ with an uptake rate
of -594 ± 152 mg m$^{-2}$ d$^{-1}$. In the pastures, $CO_2$ emissions were twice as high when dry with cumulative annual emissions of
5,748 ± 303 g m$^{-2}$ y$^{-1}$ compared to when wet, with 2,163 ± 465 g m$^{-2}$ y$^{-1}$. For the coastal wetlands, cumulative annual $CO_2$
emissions were highest in freshwater tidal forests with 2,213 ± 284 g m$^{-2}$ y$^{-1}$, followed by mangroves with 1,493 ± 111g m$^{-2}$ y$^{-1}$
and lowest at the saltmarsh with uptake rates of -264 ± 29 g m$^{-2}$ y$^{-1}$.

195        For $CH_4$ fluxes, significant differences were observed among sites and seasons ($t$ = 182.33, $n$ =237, $p < 0.001$). The
differences between different sites were substantial, with wet pasture having significantly higher $CH_4$ emissions than any other
site, with rates ~200 times higher (Fig. 2b). For tidal coastal wetlands, emissions of $CH_4$ were highest during the wet-hot
season in all the sites except for the mangroves, which had similar emissions throughout the year (Fig. 2b). Overall, cumulative
annual $CH_4$ emissions were 209 ± 36 g m$^{-2}$ y$^{-1}$ for the wet pasture followed by mangroves (0.73 ± 0.13 g m$^{-2}$ y$^{-1}$), dry pasture
(0.15 ± 0.03 g m$^{-2}$ y$^{-1}$), freshwater tidal forest (0.14 ± 0.03 g m$^{-2}$ y$^{-1}$), saltmarsh (0.04 ± 0.01 g m$^{-2}$ y$^{-1}$), and sugarcane (-0.04 ±
0.02 g m$^{-2}$ y$^{-1}$).

        For $N_2O$ fluxes, the highest emissions (54.6 ± 9.0 mg m$^{-2}$ d$^{-1}$) were from the dry pasture in the wet-hot season, followed
by sugarcane (20.5 ± 2.7 mg m$^{-2}$ d$^{-1}$) during the hot-dry period, which coincides with the post-fertilisation months (Fig. 2c).
Overall, dry pastures had the highest cumulative annual $N_2O$ emissions (7.99 ± 2.26 g m$^{-2}$ y$^{-1}$), followed by sugarcane (2.37 ±
0.68 g m$^{-2}$ y$^{-1}$), wet pasture (1.32 ± 0.33 mg m$^{-2}$ d$^{-1}$), saltmarsh (0.33 ± 0.11 mg m$^{-2}$ d$^{-1}$), freshwater tidal forests (0.04 ± 0.0 g
m$^{-2}$ y$^{-1}$) and finally, mangroves (0.02 ± 0.04 g m$^{-2}$ y$^{-1}$). However, these differences were only significant when considering the
interaction between time of the year and site ($t$ =100.21, $n$ =237, $p < 0.001$).

210        The $CH_4$ fluxes did not vary significantly between the low and high tide within all coastal wetlands. Contrarily, for
saltmarsh, $CO_2$ was taken during the high tide (1.12 ± 0.24 g m$^{-2}$ d$^{-1}$) but emitted (0.69 ± 0.4 g m$^{-2}$ d$^{-1}$) during the low tide ($F_{1,28}$
= 20.06, $p < 0.001$). Finally, for $N_2O$, fluxes differed in all coastal wetlands, with higher uptakes in the high tide for mangroves
($F_{1,28}$ = 38.28, $p < 0.001$; $F_{1,28}$ = 13.53, $p = 0.001$) and higher emissions in saltmarsh ($F_{1,28}$ = 38.31, $p < 0.001$) during low tide
(Table S4). These results suggested that there was a likely variability for $CO_2$ and $N_2O$ fluxes, depending on the time of
sampling.

The wet pasture had the highest total cumulative soil GHG emissions ($CH_4 + N_2O$) with 56,124 $CO_2$eq kg ha$^{-1}$ y$^{-1}$ followed by dry pasture 23,890 $CO_2$eq kg ha$^{-1}$ y$^{-1}$ and sugarcane 7,142 $CO_2$eq kg ha$^{-1}$ y$^{-1}$. While coastal wetlands had comparatively lower cumulative soil GHG emissions with 884, 235 and 144 $CO_2$eq kg ha$^{-1}$ y$^{-1}$ for saltmarsh, mangroves and freshwater tidal forests, respectively. Overall, the three coastal wetlands measured in this study had lower total cumulative GHG emissions at 1,263 $CO_{2\text{-eq}}$ kg ha$^{-1}$ yr$^{-1}$ compared to the alternate agricultural land uses, which emitted 87,156 $CO_{2\text{-eq}}$ kg ha$^{-1}$ yr$^{-1}$.

### 3.3 Greenhouse gas emissions and environmental factors

Overall, we found that not one single parameter measured in this could explain GHG fluxes for all sites except land-use. The $CO_2$ emissions were not significantly correlated with bulk density ($R^2 = 0.026$, $p = 0.918$, $n = 18$), % WFPS ($R^2 = -0.003$, $p = 0.99$, $n = 18$), or soil temperature ($R^2 = 0.296$, $p = 0.233$, $n = 18$). Soil $CH_4$ emissions were neither correlated with bulk density ($R^2 = -0.096$, $p = 0.706$, $n = 18$), % WFPS ($R^2 = 0\text{-}.224$, $p = 0.372$, $n = 18$) or soil temperature ($R^2 = 0.286$, $p = 0.25$, $n = 18$). Finally, no correlation was found between $N_2O$ emissions and bulk density ($R^2 = -0.349$, $p = 0.156$, $n = 18$), % WFPS ($R^2 = -0.34$, $p = 0.168$, $n = 18$), or soil temperature ($R^2 = -0.241$, $p = 0.335$, $n = 18$). Full raw dataset of GHG fluxes were provided in Table S1.

Table 2. Physicochemical characteristics for the soil of natural coastal wetlands, sugarcane and pastures (dry and ponded) for the top 30 cm of soil in tropical Australia. C= carbon, N = Nitrogen, EC = Electrical Conductivity. Values are mean ± standard error ($n = 5$).

| Site | | Depth (cm) | Gravimetric moisture content (%) | pH | EC ($\mu s\ cm^{-1}$) | Bulk density ($g\ cm^{-3}$) | %C | %N |
|---|---|---|---|---|---|---|---|---|
| Mangroves | | 0-10 | 41.7 ± 1.1 | 5.9 ± 0.1 | 12,550 ± 524 | 1.14 ± 0.05 | 2.3 ± 0.1 | 0.18 ± 0.01 |
| | | 10-20 | 34.6 ± 0.7 | 5.9 ± 0.3 | 12,164 ± 5,560 | 1.34 ± 0.03 | 1.7 ± 0.2 | 0.12 ± 0.01 |
| | | 20-30 | 31.3 ± 0.6 | 6.2 ± 0.1 | 5,560 ± 365 | 1.95 ± 0.12 | 0.9 ± 0.1 | 0.07 ± 0.01 |
| | | Mean | 35.9 ± 1.2 | 6.0 ± 0.1 | 8,930 ± 790 | 1.48 ± 0.10 | 1.6 ± 0.2 | 0.12 ± 0.01 |
| Saltmarsh | | 0-10 | 25.6 ± 1.2 | 5.8 ± 0.2 | 8,442 ± 435 | 1.12 ± 0.04 | 1.4 ± 0.1 | 0.11 ± 0.01 |
| | | 10-20 | 26.6 ± 0.3 | 5.8 ± 0.1 | 8,666 ± 437 | 1.47 ± 0.05 | 1.3 ± 0.1 | 0.12 ± 0.01 |
| | | 20-30 | 26.4 ± 0.2 | 5.9 ± 0.3 | 7,040 ± 316 | 1.56 ± 0.03 | 1.0 ± 0.3 | 0.10 ± 0.02 |
| | | Mean | 26.2 ± 0.4 | 5.8 ± 0.1 | 8,049 ± 276 | 1.38 ± 0.06 | 1.2 ± 0.1 | 0.11 ± 0.01 |
| Freshwater tidal forest | | 0-10 | 33.4 ± 0.5 | 4.4 ± 0.2 | 1,099 ± 17 | 0.46 ± 0.05 | 7.8 ± 0.1 | 0.62 ± 0.03 |
| | | 10-20 | 24.9 ± 0.6 | 4.2 ± 0.0 | 1,272 ± 164 | 0.71 ± 0.02 | 5.4 ± 0.0 | 0.46 ± 0.04 |
| | | 20-30 | 22.4 ± 0.7 | 4.2 ± 0.1 | 1,882 ± 47 | 0.83 ± 0.03 | 2.2 ± 0.1 | 0.10 ± 0.00 |
| | | Mean | 26.9 ± 1.3 | 4.3 ± 0.1 | 1,418 ± 104 | 0.59 ± 0.05 | 5.1 ± 0.6 | 0.39 ± 0.06 |
| Sugarcane | | 0-10 | 9.1 ± 0.4 | 5.7 ± 0.1 | 429 ± 12 | 1.35 ± 0.08 | 1.5 ± 0.1 | 0.10 ± 0.00 |
| | | 10-20 | 12.1 ± 0.6 | 5.3 ± 0.3 | 365 ± 11 | 1.46 ± 0.06 | 1.5 ± 0.1 | 0.12 ± 0.01 |
| | | 20-30 | 13.7 ± 0.2 | 4.7 ± 0.2 | 351 ± 2 | 1.64 ± 0.10 | 1.3 ± 0.1 | 0.10 ± 0.00 |
| | | Mean | 11.7 ± 0.6 | 5.2 ± 0.2 | 382 ± 11 | 1.48 ± 0.05 | 1.4 ± 0.1 | 0.11 ± 0.00 |
| Pasture | Dry | 0-10 | 12.4 ± 0.3 | 4.1 ± 0.0 | 378 ± 21 | 0.78 ± 0.06 | 3.1 ± 0.3 | 0.26 ± 0.03 |
| | | 10-20 | 13.6 ± 0.1 | 4.4 ± 0.1 | 279 ± 60 | 1.21 ± 0.14 | 1.6 ± 0.4 | 0.12 ± 0.04 |
| | | 20-30 | 14.5 ± 0.7 | 4.4 ± 0.3 | 84 ± 4 | 1.32 ± 0.19 | 1.6 ± 0.2 | 0.12 ± 0.02 |
| | | Mean | 13.5 ± 0.3 | 4.3 ± 0.1 | 247 ± 38 | 1.10 ± 0.10 | 2.1 ± 0.3 | 0.17 ± 0.02 |
| | Wet | 0-10 | 52.1 ± 0.4 | 4.8 ± 0.0 | 358 ± 71 | 0.62 ± 0.06 | 3.6 ± 0.3 | 0.29 ± 0.02 |
| | | 10-20 | 47.7 ± 0.4 | 4.9 ± 0.1 | 117 ± 11 | 1.30 ± 0.02 | 1.7 ± 0.1 | 0.10 ± 0.01 |
| | | 20-30 | 46.4 ± 0.2 | 5.1 ± 0.1 | 95 ± 6 | 1.31 ± 0.02 | 1.5 ± 0.1 | 0.10 ± 0.00 |
| | | Mean | 48.7 ± 0.7 | 4.9 ± 0.0 | 190 ± 39 | 1.07 ± 0.09 | 2.3 ± 0.3 | 0.16 ± 0.03 |


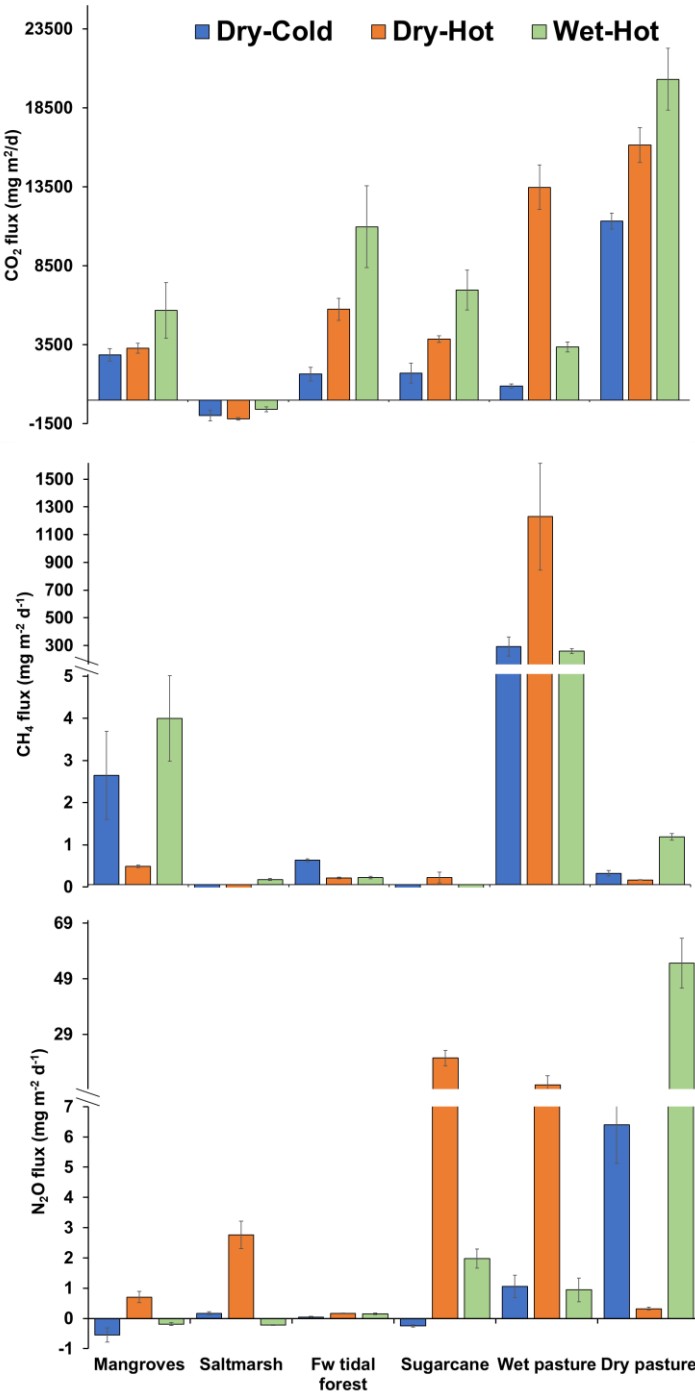

Figure 2: Greenhouse gas fluxes of (mg m$^{-2}$ d$^{-1}$) (a) $CO_2$ (b) $CH_4$ and (c) $N_2O$ from the soils of tropical coastal wetlands: mangroves, saltmarsh, freshwater (Fw) tidal forest, and two alternative land uses: sugarcane and pastures (wet and dry) during three periods of the year: dry-cold, dry-hot and wet-hot


## 4 Discussion

The soils of the three coastal wetlands measured in this study (mangroves, saltmarshes and freshwater tidal forests) had significantly lower GHG emissions than those from two alternative land uses common in tropical Australia (sugar cane and grazing pastures). Notably, we found that coastal wetlands had 200 times lower $CH_4$ emissions and seven times lower $N_2O$ compared to wet pastures and sugarcane soils, respectively. These results support our hypothesis that the management or conversion of unused sugarcane land and ponded pastures in tropical Australia could be restored to coastal wetlands and result in significant GHG mitigation.

The variability of GHG fluxes was best explained by land use and wetland type; however, some trends with seasons were evident. For instance, $CO_2$ and $N_2O$ emissions were lowest during the dry-cool periods. Reduced emissions at low temperatures are expected as the temperature is a primary driver of any metabolic process, including respiration and nitrification-denitrification. Mangroves tend to have higher $CO_2$ emissions as temperature increases (Liu and Lai, 2019), and forests have significantly higher $N_2O$ emissions during warm seasons (Schindlbacher et al., 2004). Emissions of $CH_4$ also tend to increase with temperature as the activity of soil methane-producing microbes (Ding et al., 2004) and the availability of carbon is higher in warmer conditions (Yvon-Durocher et al., 2011). However, most studies to date on GHG fluxes have been conducted in temperate locations, where temperature differences throughout the year are smaller than in tropical regions. For tropical regions, increased GHG emissions are likely to be strongly affected by the "Birch effect", which refers to short-term but a substantial increase of respiration from soils under the effect of precipitation during the early wet season (Fernandez-Bou et al., 2020).

The main factor associated with GHG fluxes was land use and type of wetland. Notably, coastal wetlands, even the freshwater tidal forests, had much lower emissions than wet pastures. This significant difference could be attributed to terminal electron acceptors in the soils (e.g. iron, sulphate, manganese) of the coastal wetlands, which could inhibit methanogenesis (Kögel-Knabner et al., 2010; Sahrawat, 2004). Sulphate reducing bacteria are also likely to outcompete methane-producing bacteria (methanogens) in the presence of high sulphate concentrations in tidal wetlands, resulting in low $CH_4$ production. Competition between methanogens and methanotrophs may result in a net balance of low $CH_4$ production despite freshwater conditions (Maietta et al., 2020). Additionally, microorganisms living within the bark of *Melaleuca* trees can consume $CH_4$ (Jeffrey et al., 2021), so it is possible that similar bacteria could reduce $CH_4$ emissions in the soil. Interestingly, variability within $CH_4$ fluxes among sites was very high, despite being very close to each other (Fig. 1b). These differences highlight the importance of land use in GHG fluxes, which are likely to significantly alter the microbial community composition and abundance, changing rapidly over small spatial scales (Drenovsky et al., 2010; Martiny et al., 2006).

Our results are consistent with other studies, showing the importance of land use in GHG emissions. For instance, in a Mediterranean climate, the drained agricultural land use types, pasture and corn, were larger $CO_2$ emitters than restored wetlands(Knox et al., 2015). Clearing of wetlands for agricultural development, such as the drainage of peatlands, results in very high $CO_2$ emissions (Hirano et al., 2012; Nieveen et al., 2005; Veenendaal et al., 2007), and restoration of these wetlands could decrease these emissions(Cameron et al., 2021). Additionally, some of the wetland types, such as marshes, were occasional sinks of $CO_2$ and $CH_4$, consistent with previous studies where intertidal wetlands sink of GHG at least under some conditions or during some times of the year (Knox et al., 2015; Maher et al., 2016).

The fluxes measured in the coastal wetlands of this study, -1191 to 10,970 mg $m^{-2}$ $d^{-1}$ for $CO_2$, -0.3 to 3.9 mg $m^{-2}$ $d^{-1}$ for $CH_4$, and -0.2 to 2.8 mg $m^{-2}$ $d^{-1}$ for $N_2O$, were within the range of those measured in other subtropical/tropical wetlands, worldwide (except for the negative $CO_2$ fluxes in saltmarsh soils, Table 3). For $CO_2$, fluxes can range between 44 and 11,328 mg $m^{-2}$ $d^{-1}$, for $CH_4$, from 0.03 to 1255 mg $m^{-2}$ $d^{-1}$ and for $N_2O$, from 0.1 to 279 mg $m^{-2}$ $d^{-1}$ (Table 3). Despite being in tropical regions, GHG fluxes from this study were lower compared to other climates (Table 3). Contrary to previous studies, $CO_2$ uptake by saltmarsh soil was likely to be linked with dark $CO_2$ fixation in wetland soils (Akinyede et al., 2020; Mar Lynn et al., 2017). Wetland soils exhibit autotrophic bacteria which contribute to dark $CO_2$ fixation at ~311 mg $m^{-2}$ $d^{-1}$ however these rates could vary depending upon abundance and diversity of microbial communities ((Akinyede et al., 2020). Further studies exploring presence and abundance of $CO_2$ fixing bacteria in saltmarsh soils is recommended. The general lower nitrogen pollution in Australia's soils and waterways compared to other countries may partially explain the lower emissions. However, the GHG flux measurements from this study did not account for the effects of vegetation, which can alter fluxes. For instance, some plant species of rice paddies (Timilsina et al., 2020) and *Miscanthus sinensis* (Lenhart et al., 2019) can increase $N_2O$ emissions, and some tree species can facilitate $CH_4$ efflux from the soil (Pangala et al., 2013). Finally, changes in emissions between low and high tides were detected for $CO_2$ and $N_2O$. Thus, future studies that include vegetation and changes within tidal cycles will improve GHG flux estimates for coastal wetlands.

Table: 3. Comparison of GHG fluxes (mg m$^{-2}$ d$^{-1}$) with other studies

| Reference | Climate | Country | Ecosystem | $CO_2$ fluxes | $CH_4$ fluxes | $N_2O$ fluxes |
|---|---|---|---|---|---|---|
| Allen et al., 2011 | Subtropical | Australia | Mangroves estuary | - | 1.5-51 | - |
| (Cabezas et al., 2018) | Subtropical | USA | Mangroves estuary | - | 0.3-2.2 | - |
| Li and Mitsch, 2016 | Subtropical | USA | Flooded brackish marsh | - | 212 ± 51 | - |
| Morse, Ardón and Bernhardt, 2012 | Subtropical | USA | Forested wetlands | 7224-11328 | 118-1255 | 46-279 |
| Musenze et al., 2014 | Subtropical | Australia | Mangroves estuary | - | 5-448 | 0.1-3.4 |
| Whiting and Chanton, 2001 | Subtropical | USA | Typha marsh | 409-477 | 189-264 | - |
| Mitsch et al., 201 | Tropical | South Africa | Seasonally flooded wetland | - | 264±29 | - |
| Krithika et al., 2008 | Tropical | India | Mangroves | - | 25-50 | - |
| Kristensen et al., 2008 | Tropical | Tanzania | Mangroves | 44-3521 | 1.9-6.5 | - |
| Biswas et al., 2007 | Tropical | India | Mangroves estuary | - | 0.03-2.16 | - |
| Purvaja et al., 2004 | Tropical | India | Mangroves estuary | - | 10-85 | - |
| Kreuzwieser et al., 2003 | Tropical | Australia | Mangroves | - | 0.6-11 | - |

| Reference | Climate | Country | Ecosystem | | | |
|---|---|---|---|---|---|---|
| Kiese and Butterbach-Bahl, 2002 | Tropical | Australia | Tropical rain forest | 2208-3288 | | 1.9-3.2 |
| Purvaja and Ramesh, 2000 | Tropical | India | Mangroves | - | 63-434 | - |
| Sotomayor et al. 1994 | Tropical | Puerto Rica | Mangroves | - | 5-110 | - |
| Barnes et al., 2006 | Tropical | India | Mangroves | - | 9-15 | - |
| Melling et al. 2012 | Tropical | Malaysia | Peat swamp forest | 3384 | 21-29 | |
| This study | Tropical | Australia | Freshwater tidal forest | 1640-10970 | 0.16-0.59 | -0.19-0.7 |
| This study | Tropical | Australia | Saltmarsh | -594-(-1191) | -0.25-0.12 | -0.22-2.76 |
| This study | Tropical | Australia | Mangroves | 2852-5669 | 0.44-3.95 | 0.04-0.16 |
| Oertel et al., 2016 | (Sub) Tropical | Global | Wetlands | - | -1.08-1169 | - |
| Oertel et al., 2016 | Temperate | Global | Wetlands | - | -1.49-1510 | - |
| Oertel et al., 2016 | Mediterranean | Global | Wetlands | - | - | -2.6-9.4 |
| Al-Haj and Fulweiler, 2020 | - | Global | Mangroves | - | -1.1-1169 | -0.2-6.3 |
| Al-Haj and Fulweiler, 2020 | - | Global | Saltmarshes | 6844-34983 | 0.38-3002 | -7.39-28.52 |
| Rosentreter et al., 2021 | - | Global | Mangroves | 4563-30800 | -0.69-10.78 | -1.69-4.65 |
| Rosentreter et al., 2021 | - | Global | Saltmarshes | 3802-20914 | 107-168 | 4.96 |
| IPCC 2013 | Tropical | Global | Swamp forests | | 30.76-2149 | |

Note: Hyphen means no data was available; GHG fluxes as $CO_2$-C, $CH_4$-C and $N_2O$-N were multiplied by 3.66, 1.34 and 1.57 respectively to calculate $CO_2$, $CH_4$ and $N_2O$ fluxes (National Greenhouse Accounts Factors, Australian Government Department of Industry, Science, Energy and Resources. 2020).

### 4.1 Management implications

Under the Paris Agreement, Australia has committed to reducing GHG emissions 26 - 28% below its 2005 levels by 2030. With annual emissions of 153 million tonnes of carbon dioxide equivalent (Mt $CO_{2\text{-eq}}$ $y^{-1}$), Queensland is a major GHG emitter in Australia (~ 28.7% of the total in 2016; DES, 2016). Of these emissions, about 18.3 Mt $CO_{2\text{-eq}}$ $y^{-1}$ (14%) are attributed to agriculture, while land-use change and forestry emit 12.1 Mt $CO_{2\text{-eq}}$ $y^{-1}$ (DES, 2016). Production of $CH_4$ from ruminant animals, primarily cattle, contribute 82% of agriculture-related emissions (DES, 2016). Therefore, any GHG mitigation strategy from
the land-use change could be important for Australia to achieve its national goals.

This study supports the application of three management actions that could reduce GHG emissions. First, the conversion of ponded pastures to coastal wetlands is likely to reduce soil GHG emissions. Our results showed that wet pastures emit 56 ton $CO_{2\text{-eq}}$ $ha^{-1}$ $y^{-1}$ of the total GHG ($CH_4$ + $N_2O$) compared with 0.2 ton $CO_{2\text{-eq}}$ $ha^{-1}$ $y^{-1}$, 0.1 ton $CO_{2\text{-eq}}$ $ha^{-1}$ $y^{-1}$ and 0.9
ton $CO_{2\text{-eq}}$ $ha^{-1}$ $y^{-1}$ from mangroves, freshwater tidal forest, and saltmarshes, respectively. This implies that about 55 ton $CO_{2\text{-eq}}$ $ha^{-1}$ $y^{-1}$ emissions from the soils could be potentially avoided by converting wet pastures to coastal wetlands. The carbon mitigation for GHG emissions only from soil could provide ~ AUD 894 $ha^{-1}$ $yr^{-1,}$ assuming a carbon value of AUD 15.99 per ton of $CO_{2\text{-eq}}$ (Australian Government Clean Energy Regulator, 2021). This mitigation could be added up to the carbon sequestration through sediment accumulation and tree growth that results from wetland restoration. Legal enablers in
Queensland are in place to manage unproductive agricultural land this way (Bell-James and Lovelock, 2019) and could provide an alternative income source for farmers.

A second management option would be to reduce the time pastures are kept underwater. Dry pastures produced
significantly less $CH_4$ with ~0.005 kg $ha^{-1}$ $d^{-1}$ than wet pastures with 6 kg $ha^{-1}$ $d^{-1}$. For comparison, an average cow produces 141 g $CH_4$ $d^{-1}$ (McGinn et al., 2011), and our study farm supported around 900 cattle over 250 ha throughout the year, equivalent to185 kg $ha^{-1}$ $y^{-1}$ compared to 2 kg $ha^{-1}$ $y^{-1}$ and 2,090 kg $ha^{-1}$ $y^{-1}$ $CH_4$ from dry and wet pasture respectively. This implies that nearly 92% of the $CH_4$ emissions came from wet pastures, while dry pasture and grazing cattle had a low share in total $CH_4$ emissions in this case scenario. Therefore, land use management of wet pastures may be an opportunity to reduce
agriculture-related $CH_4$ emissions. Future studies should increase the number of sites of ponded pastures to account for variability in hydrology, fertilisation, and on-farm cattle density. However, the exceptionally large difference (2-3 orders of

magnitude) between dry and ponded pastures and other coastal wetlands provides confidence that pasture management could provide significant GHG mitigation.

Finally, fertiliser management in sugarcane could reduce $N_2O$ emissions. Higher $N_2O$ emissions of 17.6 mg m$^{-2}$ d$^{-1}$ were measured in the sugarcane crop following fertilisation during the dry-hot season. Comparatively, natural wetlands had low $N_2O$ emissions (0.16 to 2.79 mg m$^{-2}$ d$^{-1}$), and even the saltmarsh was an occasional sink. Thus, improved management of fertiliser applications could result in GHG emission mitigation. Some activities include split application of nitrogen fertiliser in combination with low irrigation, reduction in fertiliser application rates, the substitution of nitrate-based fertiliser for urea
(Rezaei Rashti et al., 2015), removing mulch layer before fertiliser application (Pinheiro et al., 2019; Xu et al., 2019; Zaehle and Dalmonech, 2011) or conversion of unproductive sugarcane to coastal wetlands.

**5 Conclusion**

The GHG emissions from three coastal wetlands in tropical Australia (mangroves, saltmarsh and freshwater tidal forests) were consistently lower than those from two common agricultural land use of the region (sugarcane and pastures) throughout three
climatic conditions (dry-cool, dry-hot and wet-hot). Ponded pastures emitted 200 times more $CH_4$, and sugarcane emitted seven times more than any natural coastal wetland measured. If these high emissions are persistent in other locations and within other tropical regions, conversion of pastures and sugarcane to similar coastal wetlands could provide significant GHG mitigation. As nations try to reach their emission reduction targets, projects aimed at converting or restoring coastal wetlands can financially benefit farmers while providing additional co-benefits


**Author contribution**

Iram, N. and M.F. Adame designed the project, Iram, N, B. Shahrabi Farahani and E. Kavehei carried out experiments, Iram, N., E. Kavehei and M.F. Adame analysed the data. Iram, N and M.F. Adame prepared the manuscript with contributions from
D.T. Maher, S.E., Bunn, and M. Rezaei Rashti.

**Competing interests**

The authors declare that they have no conflict of interest.

**Acknowledgements**

We acknowledge the Traditional Owners of the land in which the field study was conducted, especially the Nywaigi people from Mungalla Station, where this study was conducted. We are also thankful to Sam and Santo Lamari for allowing us to work on their property and for sharing their local knowledge. We are thankful to Charles Cadier and Julieta Gamboa for their contribution to the fieldwork. This project was financially supported by an Advance Queensland Industry Research Fellowship

to MF Adame. We are very thankful to anonymous referees for their valuable feedback, which helped to improve the quality of the article.

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
