# Peer review of "Soil greenhouse gas fluxes from tropical coastal wetlands and alternative agricultural land uses"

_Biogeosciences, 2021_

## Author Comment (AC2)

**Greenhouse gas emissions from tropical coastal wetlands and their alternative agricultural lands: Where significant mitigation gains lie**

Supplementary data file 2_ Linearity test results for Dry-hot season

Iram et al. 2021

Correspondence: naima.iram@griffithuni.edu.au

**Mangroves- CO2**

**Mangroves- CH4**

$R^2 = 0.6056$

$R^2 = 0.9999$

$R^2 = 0.7983$

**Mangroves- N2O**

$R^2 = 0.7627$

$R^2 = 0.7462$

$R^2 = 0.9902$

$R^2 = 0.8041$

$R^2 = 0.7541$

$R^2 = 0.9279$

R² = 0.7211

R² = 0.9994

R² = 0.6323

**Saltmarsh- CO2**

R² = 0.8828

R² = 0.9494

R² = 0.6027

R² = 0.8355

R² = 0.9352

R² = 0.8863

$R^2 = 0.9276$

$R^2 = 0.9909$

$R^2 = 0.9521$

**Saltmarsh- CH4**

$R^2 = 0.8325$

$R^2 = 0.7307$

$R^2 = 0.7417$

$R^2 = 0.9842$

$R^2 = 0.6109$

$R^2 = 0.9683$

R² = 0.9538

R² = 0.7856

R² = 0.7354

**Saltmarsh- N2O**

R² = 0.9729

R² = 0.9924

R² = 0.6987

$R^2 = 0.96$

$R^2 = 0.997$

$R^2 = 0.8714$

$R^2 = 0.8323$

$R^2 = 0.9779$

$R^2 = 0.9854$

**Freshwater tidal forest-CO2**

$R^2 = 0.9435$

$R^2 = 0.9333$

$R^2 = 0.9481$

**Freshwater tidal forest-CH4**

**Freshwater tidal forest-N2O**

**Sugarcane- CO2**

$R^2 = 0.9606$

$R^2 = 0.8804$

$R^2 = 0.9997$

$R^2 = 0.9854$

$R^2 = 0.9904$

$R^2 = 0.8218$

**Sugarcane- CH4**

$R^2 = 0.9434$

$R^2 = 0.6964$

$R^2 = 0.8686$

$R^2 = 0.9845$

$R^2 = 0.9767$

$R^2 = 0.9866$

$R^2 = 0.9643$

$R^2 = 0.9949$

$R^2 = 0.9447$

**Sugarcane- N2O**

$R^2 = 0.9779$

$R^2 = 0.8798$

$R^2 = 0.9856$

**Wet ponded pasture-CO2**

**Wet ponded pasture-CH4**

$R^2 = 0.7568$

$R^2 = 0.918$

$R^2 = 0.7823$

$R^2 = 0.8661$

$R^2 = 0.9653$

$R^2 = 0.6357$

$R^2 = 0.9442$

$R^2 = 0.9999$

$R^2 = 0.9443$

$R^2 = 0.9985$

$R^2 = 0.9932$

**Wet ponded pasture-N2O**

**Dry ponded pasture-CO2**

**Dry ponded pasture-CH4**

**Dry ponded pasture-N2O**

**Greenhouse gas emissions from tropical coastal wetlands and their alternative agricultural lands: Where significant mitigation gains lie**

Iram et al. 2021
Correspondence to: Naima Iram (naima.iram@griffithuni.edu.au)

40 years climate data on mean maximum temperature and rainfall from Bureau of Metrology Australia,

Monthly Climate Statistics for 'LUCINDA POINT' [032141]       Source       http://www.bom.gov.au/jsp/ncc/cdio/cvg/av
Created on [ 05 May 2021 14:44:52 GMT+00:00]

032141 LUCINDA POINT
Commenced: 1980
Last Record: 2021
Latitude: 18.52 Degrees South
Longitude: 146.39 Degrees East
Elevation: 10 m
State: QLD

| Statistic Element | January | February | March | April | May | June | July | August | September | October | November | December | Annual | Number of | Start Year | End Year |
|---|---|---|---|---|---|---|---|---|---|---|---|---|---|---|---|
| Mean maximum temperature (Degrees C) for years 1980 to 2021 | 30.1 | 30.3 | 29.7 | 27.9 | 25.9 | 23.8 | 23.1 | 23.9 | 25.8 | 27 | 28.4 | 29.6 | 27.1 | 32 | 1980 | 2021 |
| Highest temperature (Degrees C) for years 1980 to 2021 | 37 | 38.5 | 35.9 | 33.1 | 31 | 29.9 | 28.4 | 29.5 | 32.5 | 31 | 34.4 | 37 | 38.5 | 32 | 1980 | 2021 |
| Date of Highest temperature for years 1980 to 2021 | 7-Jan-94 | 20-Feb-19 | 1-Mar-81 | 14-Apr-81 | 12-May-92 | 1-Jun-81 | 27-Jul-98 | 18-Aug-81 | 22-Sep-86 | 20-Oct-86 | 28-Nov-18 | 6-Dec-91 | 20-Feb-19 | N/A | 1980 | 2021 |
| Lowest maximum temperature (Degrees C) for years 1980 to 2021 | 25.2 | 24.7 | 25.5 | 23.2 | 17.3 | 17.8 | 17.5 | 19.4 | 22.2 | 22.6 | 23.9 | 25.8 | 17.3 | 32 | 1980 | 2021 |
| Date of Lowest maximum temperature for years 1980 to 2021 | 13-Jan-81 | 20-Feb-21 | 4-Mar-98 | 30-Apr-82 | 23-May-20 | 21-Jun-07 | 15-Jul-16 | 14-Aug-05 | 1-Sep-86 | 21-Oct-10 | 27-Nov-99 | 24-Dec-99 | 23-May-20 | N/A | 1980 | 2021 |
| Decile 1 maximum temperature (Degrees C) for years 1980 to 2021 | 28.5 | 28.2 | 27.9 | 26.2 | 24.2 | 21.8 | 21.4 | 22 | 23.9 | 25.4 | 26.9 | 28.1 | | 30 | 1980 | 2021 |
| Decile 9 maximum temperature (Degrees C) for years 1980 to 2021 | 32 | 32.3 | 31.8 | 29.9 | 28 | 25.9 | 25 | 26.1 | 28 | 28.7 | 29.9 | 31.4 | | 30 | 1980 | 2021 |
| Mean number of days >= 30 Degrees C for years 1980 to 2021 | 14.3 | 14 | 10.6 | 2.2 | 0.1 | 0 | 0 | 0 | 0.2 | 0.2 | 2.5 | 10 | 54.1 | 32 | 1980 | 2021 |
| Mean number of days >= 35 Degrees C for years 1980 to 2021 | 0.1 | 0.4 | 0.1 | 0 | 0 | 0 | 0 | 0 | 0 | 0 | 0 | 0.1 | 0.7 | 32 | 1980 | 2021 |
| Mean number of days >= 40 Degrees C for years 1980 to 2021 | 0 | 0 | 0 | 0 | 0 | 0 | 0 | 0 | 0 | 0 | 0 | 0 | 0 | 32 | 1980 | 2021 |
| Mean minimum temperature (Degrees C) for years 1980 to 2021 | 25.3 | 25.2 | 24.8 | 23.4 | 21.3 | 19 | 18.1 | 18.9 | 20.9 | 22.9 | 24.2 | 25.3 | 22.4 | 32 | 1980 | 2021 |
| Lowest temperature (Degrees C) for years 1980 to 2021 | 19.7 | 18.7 | 21 | 16.7 | 13.9 | 11.7 | 11.2 | 12.5 | 13 | 17 | 20.4 | 20.3 | 11.2 | 32 | 1980 | 2021 |
| Date of Lowest temperature for years 1980 to 2021 | 25-Jan-86 | 4-Feb-15 | 12-Mar-03 | 24-Apr-03 | 24-May-20 | 24-Jun-82 | 20-Jul-85 | 2-Aug-03 | 26-Sep-96 | 6-Oct-93 | 4-Nov-09 | 24-Dec-85 | 20-Jul-85 | N/A | 1980 | 2021 |
| Highest minimum temperature (Degrees C) for years 1980 to 2021 | 29 | 29.2 | 28.8 | 27.3 | 26 | 23.6 | 23.1 | 23.6 | 25.4 | 26.4 | 28.5 | 29.1 | 29.2 | 32 | 1980 | 2021 |
| Date of Highest minimum temperature for years 1980 to 2021 | 11-Jan-83 | 14-Feb-17 | 30-Mar-17 | 8-Apr-83 | 8-May-96 | 15-Jun-20 | 21-Jul-98 | 22-Aug-10 | 29-Sep-16 | 27-Oct-05 | 28-Nov-18 | 20-Dec-95 | 14-Feb-17 | N/A | 1980 | 2021 |
| Decile 1 minimum temperature (Degrees C) for years 1980 to 2021 | 23.5 | 23.3 | 23 | 21.4 | 19 | 15.9 | 14.7 | 16.6 | 18.8 | 20.6 | 22.2 | 23.3 | | 30 | 1980 | 2021 |
| Decile 9 minimum temperature (Degrees C) for years 1980 to 2021 | 27.2 | 27 | 26.6 | 25.4 | 23.4 | 21.6 | 20.7 | 21 | 23 | 24.8 | 26 | 27.1 | | 30 | 1980 | 2021 |
| Mean number of days <= 2 Degrees C for years 1980 to 2021 | 0 | 0 | 0 | 0 | 0 | 0 | 0 | 0 | 0 | 0 | 0 | 0 | 0 | 32 | 1980 | 2021 |
| Mean number of days <= 0 Degrees C for years 1980 to 2021 | 0 | 0 | 0 | 0 | 0 | 0 | 0 | 0 | 0 | 0 | 0 | 0 | 0 | 32 | 1980 | 2021 |
| Mean daily ground minimum temperature Degrees C for years null to null | | | | | | | | | | | | | | | | |
| Lowest ground temperature Degrees C for years null to null | | | | | | | | | | | | | | | | |
| Date of Lowest ground temperature  for years null to null | | | | | | | | | | | | | N/A | | | |
| Mean number of days ground min. temp. <= -1 Degrees C for years null to null | | | | | | | | | | | | | | | | |
| Mean rainfall (mm) for years 1980 to 2020 | 191.7 | 210 | 176 | 89.7 | 60.1 | 35.2 | 22.1 | 23.8 | 24.9 | 25.9 | 75.2 | 99.7 | 1076.1 | 29 | 1980 | 2020 |
| Highest rainfall (mm) for years 1980 to 2020 | 880.8 | 542.8 | 684 | 312 | 195.6 | 194.4 | 166.8 | 139.4 | 151.4 | 223.6 | 607.4 | 568.8 | 2208.8 | 34 | 1980 | 2020 |
| Date of Highest rainfall for years 1980 to 2020 | 1981 | 2009 | 2016 | 1990 | 1981 | 1990 | 1981 | 1997 | 2010 | 2010 | 2010 | 2018 | 2010 | N/A | 1980 | 2020 |
| Lowest rainfall (mm) for years 1980 to 2020 | 0 | 32 | 35.6 | 8.6 | 1.4 | 2.8 | 0 | 0 | 0 | 0 | 0 | 0 | 504.9 | 34 | 1980 | 2020 |
| Date of Lowest rainfall for years 1980 to 2020 | 2002 | 2011 | 2008 | 1994 | 2001 | 2004 | 1995 | 2018 | 1991 | 2001 | 2019 | 2001 | 2002 | N/A | 1980 | 2020 |
| Decile 1 monthly rainfall (mm) for years 1980 to 2020 | 21.5 | 44.2 | 42.3 | 22.5 | 12 | 8.2 | 4.2 | 0.7 | 0.5 | 1 | 3.6 | 16.2 | 593 | 34 | 1980 | 2020 |
| Decile 5 (median) monthly rainfall (mm) for years 1980 to 2020 | 168.6 | 176.6 | 133.2 | 80.8 | 56.8 | 25.6 | 15.5 | 15.2 | 16.2 | 11.2 | 27.2 | 76.2 | 1047 | 34 | 1980 | 2020 |
| Decile 9 monthly rainfall (mm) for years 1980 to 2020 | 314.1 | 412.9 | 315.6 | 148.6 | 97.2 | 64.2 | 40.5 | 50.7 | 53.8 | 59 | 216.4 | 151.8 | 1605.2 | 34 | 1980 | 2020 |
| Highest daily rainfall (mm) for years 1980 to 2020 | 173.6 | 213 | 175 | 134 | 94.4 | 167 | 134 | 81 | 65.4 | 106.2 | 132 | 390.4 | 390.4 | 32 | 1980 | 2020 |
| Date of Highest daily rainfall for years 1980 to 2020 | 17-Jan-81 | 25-Feb-18 | 17-Mar-00 | 11-Apr-06 | 21-May-81 | 6-Jun-90 | 25-Jul-81 | 31-Aug-98 | 20-Sep-10 | 21-Oct-10 | 2-Nov-13 | 16-Dec-18 | 16-Dec-18 | N/A | 1980 | 2020 |
| Mean number of days of rain for years 1980 to 2020 | 15.1 | 14.4 | 15.1 | 14 | 12 | 7.4 | 6.7 | 6.4 | 5.2 | 5.6 | 7.9 | 10.5 | 120.3 | 34 | 1980 | 2020 |
| Mean number of days of rain >= 1 mm for years 1980 to 2020 | 11.7 | 11.3 | 11.6 | 9.6 | 7.5 | 4.7 | 3.4 | 3.1 | 2.7 | 3.4 | 5 | 7 | 81 | 32 | 1980 | 2020 |
| Mean number of days of rain >= 10 mm for years 1980 to 2020 | 5.3 | 5.2 | 4.4 | 2.5 | 1.7 | 0.8 | 0.4 | 0.6 | 0.8 | 0.6 | 2 | 2.3 | 26.6 | 32 | 1980 | 2020 |
| Mean number of days of rain >= 25 mm for years 1980 to 2020 | 2.5 | 2.5 | 2.2 | 0.8 | 0.4 | 0.2 | 0.1 | 0.2 | 0.2 | 0.2 | 0.9 | 1 | 11.2 | 32 | 1980 | 2020 |
| Mean daily wind run (km) for years 2003 to 2021 | 425 | 463 | 426 | 424 | 397 | 406 | 394 | 358 | 362 | 427 | 425 | 413 | 410 | 13 | 2003 | 2021 |
| Maximum wind gust speed (km/h) for years 1980 to 2021 | 115 | 185 | 109 | 96 | 82 | 68 | 94 | 72 | 76 | 113 | 74 | 84 | 185 | 22 | 1980 | 2021 |
| Date of Maximum wind gust speed for years 1980 to 2021 | 13-Jan-09 | 2-Feb-11 | 20-Mar-06 | 13-Apr-14 | | 11-Jun-15 | 20-Jul-05 | | | 24-Oct-05 | | 2-Feb-11 | N/A | | 1980 | 2021 |
| Mean daily sunshine (hours) for years null to null | | | | | | | | | | | | | | | | |
| Mean daily solar exposure (MJ/(m*m)) for years 2007 to 2021 | 19.4 | 18.9 | 17.5 | 15.8 | 14.1 | 13.3 | 14.4 | 17.6 | 20 | 22 | 22.8 | 22.8 | 18.2 | 14 | 2007 | 2021 |
| Mean number of clear days for years null to null | | | | | | | | | | | | | | | | |
| Mean number of cloudy days for years null to null | | | | | | | | | | | | | | | | |
| Mean daily evaporation (mm) for years null to null | | | | | | | | | | | | | | | | |
| Mean 9am temperature (Degrees C) for years 1980 to 2010 | 27.5 | 27.6 | 26.8 | 25 | 22.9 | 20.7 | 19.6 | 20.5 | 22.9 | 25.1 | 26.6 | 27.6 | 24.4 | 20 | 1980 | 2010 |
| Mean 9am wet bulb temperature (Degrees C) for years 2000 to 2010 | | | | | | | | | | | | | | 7 | 2000 | 2010 |
| Mean 9am dew point temperature (Degrees C) for years 2000 to 2010 | | | | | | | | | | | | | | 7 | 2000 | 2010 |
| Mean 9am relative humidity (%) for years 2000 to 2010 | | | | | | | | | | | | | | 7 | 2000 | 2010 |
| Mean 9am cloud cover (okas) for years null to null | | | | | | | | | | | | | | | | |
| Mean 9am wind speed (km/h) for years 1980 to 2010 | 16 | 17 | 19.4 | 21.4 | 21.3 | 21.3 | 20.3 | 18.6 | 16.7 | 16.2 | 15.3 | 14.8 | 18.2 | 18 | 1980 | 2010 |
| Mean 3pm temperature (Degrees C) for years 1980 to 2010 | 28.7 | 28.9 | 28.5 | 26.8 | 24.9 | 22.8 | 22.4 | 22.7 | 24.4 | 25.8 | 27 | 28.2 | 25.9 | 20 | 1980 | 2010 |
| Mean 3pm wet bulb temperature (Degrees C) for years 2000 to 2010 | | | | | | | | | | | | | | 7 | 2000 | 2010 |
| Mean 3pm dew point temperature (Degrees C) for years 2000 to 2010 | | | | | | | | | | | | | | 7 | 2000 | 2010 |
| Mean 3pm relative humidity (%) for years 2000 to 2010 | | | | | | | | | | | | | | 7 | 2000 | 2010 |
| Mean 3pm cloud cover (oktas) for years null to null | | | | | | | | | | | | | | | | |
| Mean 3pm wind speed (km/h) for years 1980 to 2010 | 21.6 | 20.7 | 20.7 | 20.3 | 17.9 | 17.7 | 17.2 | 18.9 | 20.9 | 23.6 | 23 | 22.9 | 20.4 | 18 | 1980 | 2010 |

**Greenhouse gas emissions from tropical coastal wetlands and their alternative agricultural lands: Where significant mitigation gains lie**

*Correspondence to*: Naima Iram (naima.iram@griffithuni.edu.au)

**Difference in physiochemical parameters across seasonal cycle**

**Nonparametric Tests**

**Hypothesis Test Summary**

| | Null Hypothesis | Test | Sig.[a,b] | Decision |
|---|---|---|---|---|
| 1 | The distribution of Gravimetric water content% is the same across categories of Season . | Independent-Samples Kruskal-Wallis Test | .033 | Reject the null hypothesis. |
| 2 | The distribution of BD (g cm-3) is the same across categories of Season . | Independent-Samples Kruskal-Wallis Test | .000 | Reject the null hypothesis. |
| 3 | The distribution of Soil Tem is the same across categories of Season . | Independent-Samples Kruskal-Wallis Test | .000 | Reject the null hypothesis. |

a. The significance level is .050.

b. Asymptotic significance is displayed.

**Independent-Samples Kruskal-Wallis Test**

**Gravimetric water content% across Season**

**Independent-Samples Kruskal-Wallis Test Summary**

| | |
|---|---|
| Total N | 90 |
| Test Statistic | 6.800[a] |
| Degree Of Freedom | 2 |

| Asymptotic Sig.(2-sided test) | .033 |
|---|---|

a. The test statistic is adjusted for ties.

**Independent-Samples Kruskal-Wallis Test**

**Pairwise Comparisons of Season**

| Sample 1-Sample 2 | Test Statistic | Std. Error | Std. Test Statistic | Sig. | Adj. Sig.ᵃ |
|---|---|---|---|---|---|
| Dry-Hot-Dry-Cold | .317 | 6.743 | .047 | .963 | 1.000 |

| | | | | | |
|---|---|---|---|---|---|
| Dry-Hot-Wet-Hot | -15.383 | 6.743 | -2.281 | .023 | .068 |
| Dry-Cold-Wet-Hot | -15.067 | 6.743 | -2.234 | .025 | .076 |

Each row tests the null hypothesis that the Sample 1 and Sample 2 distributions are the same.

 Asymptotic significances (2-sided tests) are displayed. The significance level is .050.

a. Significance values have been adjusted by the Bonferroni correction for multiple tests.

**BD (g cm-3) across Season**

**Independent-Samples Kruskal-Wallis Test Summary**

| | |
|---|---|
| Total N | 90 |
| Test Statistic | 54.599[a] |
| Degree Of Freedom | 2 |
| Asymptotic Sig.(2-sided test) | .000 |

a. The test statistic is adjusted for ties.

**Independent-Samples Kruskal-Wallis Test**

**Pairwise Comparisons of Season**

| Sample 1-Sample 2 | Test Statistic | Std. Error | Std. Test Statistic | Sig. | Adj. Sig.[a] |
|---|---|---|---|---|---|
| Dry-Cold-Dry-Hot | -33.683 | 6.729 | -5.006 | .000 | .000 |
| Dry-Cold-Wet-Hot | -48.517 | 6.729 | -7.210 | .000 | .000 |
| Dry-Hot-Wet-Hot | -14.833 | 6.729 | -2.204 | .028 | .083 |

Each row tests the null hypothesis that the Sample 1 and Sample 2 distributions are the same.

Asymptotic significances (2-sided tests) are displayed. The significance level is .050.

a. Significance values have been adjusted by the Bonferroni correction for multiple tests.

**Soil Tem across Season**

**Independent-Samples Kruskal-Wallis Test Summary**

| | |
|---|---|
| Total N | 90 |
| Test Statistic | 57.561[a] |
| Degree Of Freedom | 2 |
| Asymptotic Sig.(2-sided test) | .000 |

a. The test statistic is adjusted for ties.

| Sample 1-Sample 2 | Test Statistic | Std. Error | Std. Test Statistic | Sig. | Adj. Sig.[a] |
|---|---|---|---|---|---|
| Dry-Cold-Wet-Hot | -42.583 | 6.724 | -6.333 | .000 | .000 |
| Dry-Cold-Dry-Hot | -45.617 | 6.724 | -6.784 | .000 | .000 |
| Wet-Hot-Dry-Hot | 3.033 | 6.724 | .451 | .652 | 1.000 |

Each row tests the null hypothesis that the Sample 1 and Sample 2 distributions are the same.

Asymptotic significances (2-sided tests) are displayed. The significance level is .050.

a. Significance values have been adjusted by the Bonferroni correction for multiple tests.

**Physiochemical properties across three depths**

| Parameter | $t$ | $n$ | $p$ |
|---|---|---|---|
| Gravimetric water | 0.268 | 90 | 0.875 |
| pH | 0.109 | 90 | 0.947 |
| EC | 5.134 | 90 | 0.077 |
| BD | 25.994 | 90 | 0.000 |
| C | 20.407 | 90 | 0.000 |
| N | 21.467 | 90 | 0.000 |

Table 1. Greenhouse gas (GHG) fluxes from soils of tropical coastal wetlands: mangroves, saltmarsh, and freshwater (Fw) tidal forest during high and low tide during a dry-hot season

| GHG | Land-use type | High tide Mean | SE | Low tide Mean | SE |
|---|---|---|---|---|---|
| $CO_2$ (g m$^{-2}$ d$^{-1}$) | Mangroves | 2.55 | 0.37 | 3.25 | 0.57 |
|  | Saltmarsh | -1.12 | 0.24 | 0.69 | 0.40 |
|  | FW tidal forest | 2.97 | 1.35 | 5.35 | 2.68 |
| $CH_4$ (mg m$^{-2}$ d$^{-1}$) | Mangroves | 3.38 | 0.98 | 236 | 73 |
|  | Saltmarsh | -0.13 | 0.06 | -25 | 6 |
|  | Fw tidal forest | 1.10 | 0.52 | 457 | 108 |
| $N_2O$ (mg m$^{-2}$ d$^{-1}$) | Mangroves | -0.74 | 0.17 | 0.15 | 0.06 |
|  | Saltmarsh | 0.19 | 0.06 | -0.14 | 0.04 |
|  | FW tidal forest | 0.06 | 0.01 | -0.25 | 0.16 |

---

## Author Response (AR1)

**Author's response on comments from Associate editor**

Thanks for your decision about our manuscript. Following your decision, major revisions were made in the manuscript including title modification, modulating the discussion, management implication and conclusion sections and limitations of the study were acknowledged. We also carefully checked the grammar and linguistic aspects of the manuscript and the author's response file and uploaded the revised documents.

**Author's response on comments on bg-2021-28 by anonymous refree#1/ RC2**

The study presents interesting findings of GHG measurement from wetlands, and their competing land uses expansion in Australia. I appreciate that the authors have incorporated my previous review comments, specifically by adding their raw data through SI. The current version is well improved. Please find below some specific recommendations which may be useful.

Please note that line numbers in authors response refer to line numbers in revised manuscript. And changes made in original manuscript were highlighted in yellow.

**Author's response**: We thank anonymous referee#1 for constructive feedback and for highlighting the improvement in the quality of the revised manuscript. The received recommendations were carefully considered and incorporated into the current version of the manuscript. A point-by-point response to comments was given below.

**1.** Line 40: this opening sentence sounds awkward and unfinished.

**Author's response**:

We have rewritten the Introduction, including the paragraph referred to by the reviewer:

L35: "The GHG emissions in coastal wetlands primarily result from microbial processes in the soil-water-atmosphere interface (Bauza et al, 2002; Whalen, 2005)."

**2.** Line 51: need reference.

**Author's response**:

We have added a reference (Knox et al, 2015) in L44.

L44: "Despite potential high GHG emissions from coastal wetlands, these are likely to be lower than those from alternative agricultural land uses (Knox et al., 2015), which emit GHGs from their construction throughout their productive lives."

**3.** Lines 53-54: I suggest finding alternative reference since (if I am correct) Boone's papers did not measure CO2 oxidation directly through gas sampling or analyser. They used stock changes instead, which is hard to find out the process underlying lowering soil carbon stocks.

**Author's response**:

Thanks for the correction. Alternative references were included as following:

L45: "Firstly, when wetlands are converted to agricultural land, the oxidation of sequestered carbon in the organic-rich soils release significant amounts of CO2 (Drexler, de Fontaine, & Deverel, 2009; Hooijer et al, 2012)."

**4.** Line 57: how about CH4 emissions from the artificial ditch? I see lots of artificial ditch

**Author's response**: Yes, drains can also be a source of CH4 in agricultural landscapes; we have added the following information:

L47: "Secondly, removing tidal flow and converting coastal wetlands to freshwater systems, such as during the creation of ponded pastures, dams or agricultural ditches can result in very high CH4 emissions (Deemer et al., 2016; Grinham et al, 2018; Ollivier et al, 2019). For instance, agricultural ditches contribute up to 3% of the total anthropogenic $CH_4$ emissions globally (Peacock et al., 2021)."

**5.** Line 60: …changing the balance between carbon and nitrogen…. Could you explain a bit more on this process? Any reference?

**Author's response**:

The sentence was removed from the revised introduction.

**6.** Line 77: …reinstallation of tidal inundation…, tidal flow restoration?

**Author's response**:

We have chosen the term "reinstallation of tidal flow" as it implies that there was inundation that was interrupted. We have clarified as follows:

L51: "Thus, emissions of GHG from land-use change can be mitigated through the reversal of these activities, for instance, reduction of fertiliser use and the reinstallation of tidal flow on unused agricultural land (Rashti et al, 2016; Kroeger et al. 2017)."

**7.** Lines 79-80: Tidal coastal wetlands?

**Author's response**:

This section was removed from the revised introduction.

**8.** Line 87: change information to data

**Author's response**:

This section was removed from the revised Introduction.

**9.** Line 97-103: move the current last sentence to the second.

**Author's response**:

This section was fully revised as following:

L55: "This study measured the annual GHG fluxes ($CO_2$, $CH_4$ and $N_2O$) from three natural coastal wetlands (mangroves, saltmarsh and freshwater tidal forests) and two agricultural land use sites (sugarcane plantation and pasture) in tropical Australia. The objectives were to assess the difference in GHG fluxes throughout different seasons that characterise tropical climates (dry-cool, dry-hot and wet-hot) and to identify environmental factors associated with these GHG fluxes.  These data will inform emission factors for converting wetlands to agricultural land uses and vice versa, filling in a knowledge gap identified in Australia (Baldock et al., 2021) and tropical regions worldwide (IPCC 2013)."

**10.** L105: In the study site text, I haven't seen any description about the original land cover prior to sugarcane and pasture, were they mangrove, salt marsh or tidal forest? There is still missing information on the reason behind study sites/land cover selection.

**Author's response**:

The requested information was added for clarity as follows:

L68: "Wetlands in this region were heavily deforested in the past century (1943- 1996) due to rapid agricultural development, primarily for sugarcane farming (Griggs. 2017). Before clearing, the land was mostly covered by rainforest and coastal wetlands, mainly Melaleuca forest, grass and sedge swamps (Johnson, Ebert, & Murray, 1999)."

**11.** Lines 131-137: please describe how did you measure at two different tide conditions (low vs high tide). Did you use a floating collar? Also, currently how spatial replication was performed within site is unclear. You may want to add this information in table 1.

**Author's response**:

We measured five replicate chambers per site to account for small scale variability; the differences within chambers was not statistically significant ($p > 0.05$). For the measurements at different tidal inundation levels (which were always $< 30$ cm within our sampling sites), we used the same static chambers with vertical extension to avoid full submersion. We have clarified as follows:

L132-136. "We used static, manual gas chambers made of high-density, round polyvinyl chloride pipe, which consisted of two units: a base ($r = 12$ cm, $h = 18$ cm) and a detachable collar ($h = 12$ cm; Hutchinson and Mosier, 1981; Kavehei et al, 2021). The chambers had lateral holes that could be left covered with rubber bungs at low water levels and left open at high water levels to allow for water movement between sampling events. When the wetlands were inundated for the experiments, we used PVC extensions ($h = 18$ cm)."

**12.** Figure 1: I would suggest adding sampling location points in figure 1a.

**Author's response**:

We have included the locations of the coastal wetlands within Insulator Creek as seen in Figure 1b

[Figure]

Figure 1: a) Location of sampling sites (Insulator Creek and Mungalla) in the Herbert River catchment, northeast Australia, (b) natural wetlands adjacent to sugarcane farm in Insulator Ck and sampling locations, and (c) mangroves, (d) saltmarsh, (e) freshwater tidal forest, (f) sugarcane, (g) dry ponded pasture and (h) wet ponded pasture. Pictures by N. Iram and MF Adame.

**13.** Lines 161-172: did you cut any below ground roots during collar installation? Is one day sufficient to avoid the effect of soil disturbance during collar installation? I have a particular concern about the effect of disturbance from the installation. I understand that fieldwork is always tricky. Otherwise, you could describe this as a study limitation in the discussion or provide relevant reference if required.

**Author's response**:

The chambers were installed in areas that were mostly free of roots. To avoid the effects of increased GHG emissions due to soil disturbance, we conducted three measurements at three days within a week of sampling. We did not detect any significant differences among days (p >0.05), which gives us confidence that the initial disturbance by setting the chambers was not a major cause of data discrepancy. We have clarified as follows:

L136-140: "Five chambers were set ~ 5cm deep in the soil at random locations one day before sampling to minimise the disturbance of installation during the experiment (Rashti et al, 2015). The chambers were selectively located on soil with minimal vegetation, roots, and crab burrows. We were careful not to tramp around the chambers during installation and sampling. The fact that emissions were not significantly different among days (p >0.05) provided us with confidence that disturbance due to installation was not problematic."

**14.** Line 168: did you collect two samples with 1-hour interval from each chamber? Was it sufficient to calculate flux?

**Author's response**:

To measure the linearity of the GHG fluxes over time, we collected four samples at 0, 20, 40 and 60 minutes. However, for GHG flux calculations, we collected two samples from all five chambers at 0 and 60 minutes. Our previous experience with this method has taught us that this is the most cost-effective way to measure GHG from wetlands (Kavehei et al. 2021) and agricultural lands (Rashti et al. 2015), which usually have relatively high emissions. This was described in the manuscript as following:

L142: "At the start of the experiment, gas chambers were closed. A sample was taken at time zero and then after one hour with a 20 ml syringe and transferred to a 12 mL-vacuumed exetainer (Exetainer, Labco Ltd., High Wycombe, UK). During the wet-hot season, linearity tests of GHG fluxes with time were conducted by sampling at 0, 20, 40 and 60 min (Rashti et al, 2016). For the rest of the experiments, linearity tests were performed in one of the five chambers at each site; R2 values were consistently above 0.70."

**15.** Line 188: how about the other sampling periods?

**Author's response**:

The inaccessibility of these sites during most of the year due to permission for access into farms, adverse weather during most of the year (e.g. during very hot conditions or during flooding), safety risk due to crocodiles and the high cost of sample analysis (>$AUD 8,000 per experiment) limited our replication in time and space. However, we know from our experience that temperature and rainfall are the main drivers of emissions; thus, we concentrated our

efforts in account for these two factors by including three main periods: dry-cool, wet-hot and dry-hot. We described this in the manuscript as following:

L63: "The region has a tropical climate with a mean monthly minimum temperature from 14 to 23˚C and mean monthly maximum temperature from 25 to 33˚C (Australian Bureau of Meteorology, ABM, 2020; 1968-2020; Table S3). The average rainfall is 2,158 mm y-1, with the highest values of 476 mm during February (ABM 2020; 1968-2020; Table S3)."

L88: "Each of the five sites was sampled during three periods dry-cool (May-September), dry-hot (October-December) and wet-hot (January-April; Table 1). During each time, soil physicochemical properties and GHG fluxes were measured as detailed below."

And we have also included the climatic characteristics during each of our sampling in a new Table 1:

Table 1. Mean daily air temperature and rainfall (Ingham, Qld. weather station) during sampling.

| Season | Study period | Daily min temperature (°C) | Daily max temperature (°C) | Rainfall (mm d$^{-1}$) |
|---|---|---|---|---|
| Dry-cool | 17/06/2018 | 13.4 – 14.6 | 27.7 - 28.2 | 0 |
| Dry-hot | 23-29/10/2018 | 15.7 – 21.1 | 32.2 - 36.2 | 0 |
| Dry-cool | 31/05 to 6/06/2019 | 10.9 – 17.5 | 21.6 - 28.2 | 0-25 |
| Wet-hot | 17-22/02/2020 | 23.9 – 25.3 | 33.6 - 34.5 | 0-86 |

**16.** Lines 214-215: to me, the bulk density for mangrove and salt marsh are very high, completely different than I observed in low tropics especially for mangrove. This may also reflect in very low C content as provided in Table 2.

**Author's response**:

Yes, bulk density of mangroves is comparatively higher as compared to other tropical mangroves which ranged between 0.1-.07 for top 30 cm (Adame et al., 2013) In this region,

the sediment is mostly composed of clay delivered through inundation in the floodplain, limiting the "accommodation space" to be filled by mangrove roots. As a result, the soil carbon content is not particularly high as shown in Serrano et al. 2019, C stocks and sequestration rates in Australian tropical mangroves ranged between $236\pm141$ Mg C ha$^{-1}$ year$^{-1}$ and $1.5\pm1.09$ Mg C ha$^{-1}$ year$^{-1}$ respectively.

**17.** Lines 206-219: please provide your stats results in the text, at least p-value, particularly when you compared measured variables between sites and depths.

**Author's response**: We included p values and added the analysis results file as a supplementary file (S4). We described this in the manuscript as following:

L179. "Soil physicochemical parameters (mean values 0-30 cm) varied among sites (Table 2, see full results of statistical analyses in Supplementary Material)."

**18.** Figure 2: I would suggest enlarging x-axis labels and chart bars, as well as provide statistical differences note.

Thanks for the suggestion. The figure was improved.

**19.** Table 3: please provide N sample size.

**Author's response**:

We measured five replicates from each site and reported in the manuscript as follows:

L 235 (n=5)

**20.** Lines 274-276: I am surprised that all GHGs are not correlated with temperature. How about root contribution to CO2 effluxes?

**Author's response**:

We were also expecting a stronger effect; however, when we analysed the whole dataset, the effect of land use overridden any effect of temperature or rainfall. It is also true that in tropical regions, mean temperatures do not differ so much among seasons. For example, in our study sites, the lowest and highest monthly mean temperatures were 18-25°C and 23-30°C, respectively (S3, Bureau of Meteorology, http://www.bom.gov.au/jsp/ncc/cdio/cvg/av). We have included a statement on the effect of temperature in tropical regions:

L246-254: "The variability of GHG fluxes was best explained by land use and wetland type; however, some trends with seasons were evident. For instance, $CO_2$ and $N_2O$ emissions were lowest during the dry-cool periods. Reduced emissions at low temperatures are expected as the temperature is the main driver of any metabolic process, including respiration and nitrification-denitrification. Mangroves tend to have higher $CO_2$ emissions as temperature increases (Liu and Lai 2019), and terrestrial forests have significantly higher $N_2O$ emissions during warm seasons (Schindlbacher et al, 2004). Emissions of $CH_4$ also tend to increase with temperature as the activity of soil methane-producing microbes (Ding et al, 2004) and the availability of carbon is higher in warmer conditions (Yvon-Durocher et al, 2011). However, these, as most studies in GHG fluxes, were conducted in temperate and subtropical locations where differences in temperature throughout the year are much larger than those in tropical regions."

**21.** Line 285: how did you calculate total cumulative GHG emissions? Did you use GWP? This new paper may be useful and relevant: https://link.springer.com/article/10.1007/s10021-021-00631-x

**Author's response**:

Total cumulative GHG emissions were calculated by the equation described by Shaaban et al. (2015). Thanks for suggesting a recent paper on GWP; we cited this paper to clarify the difference between GWP matrix and $CO_{2\text{-equivalent}}$ calculations. We described this in the method section as following:

L155-168: "Seasonal cumulative GHG fluxes were calculated by modifying the equation described by Shaaban et al. (2015; Eq. 2):

$$\text{Seasonal cumulative GHG fluxes} = \sum_{i=1}^{n}(\text{Ri} \times 24 \times \text{Di} \times 17.381)$$

Where;

Ri = Gas emission rate (mg m-2 hr-1 for CO2 and µg m-2 hr-1 for CH4 and N2O),

Di = number of the sampling days in a season,

17.38 = number of weeks in each period, assuming these conditions were representative of the annual cycle (see Table 1).

Annual cumulative soil GHG fluxes (CH4 + N2O) were calculated by integrating cumulative seasonal fluxes. These estimations did not account for soil CO2 values as our methodology with dark chambers only accounted for emissions from respiration and excluded uptake from primary productivity. The CO2-equivalent (CO2-eq) values were estimated by multiplying CH4 and N2O emissions by 25 and 298, respectively (IPCC 2007), which represent the radiative balance of these gases (Neubauer, 2021).

**22.** Lines 330-336: I would suggest citing the organisation name rather than website links

**Author's response**:

The suggestion was incorporated in manuscript L295-306.

**Author's responses to the comments from refree#2/ RC1**

Thank you for the invitation to review this manuscript. I have found the paper interesting and enjoyed learning about the study system. The paper is ambitious and presents management recommendations that would be of relevance to policymakers and land users. I have made comments and suggestion, which are listed below, aiming to support the authors in their ambition to offer evidence-based management solutions to coastal wetlands.

**Author's response**: We thank anonymous referee#2 for constructive feedback. The received comments were carefully considered, and revisions were made accordingly to improve the quality of the manuscript. Specifically, we have highlighted the strengths and limitations of our conclusions and how this information can guide future measurements of greenhouse gas emissions in different land uses. We acknowledge that land use replication was limited (one site per land use), but we wanted to focus on tackling small scale variation (five chambers per plot) and, importantly, temporal variation (seasonal- 3 seasons for two years). In total, we collected 237 samples in four sampling campaigns during June 2018- February 2020 that showed that land use, followed by temperature and rainfall, were affecting greenhouse gas fluxes. Future studies should aim at focusing on replication on land use.

We have included a point-by-point response to comments raised by the reviewer, and a revised manuscript has been submitted.

**1.** L26. The last part of the sentence about financial incentives does not follow logically from the first part. Please rephrase.

**Author's Response:**

We have rewritten as follows:

L24-26: "Converting unproductive sugarcane land or pastures (especially ponded ones) to coastal wetlands could provide significant GHG mitigation."

**2.** L34-36 Clunk sentence, please rephrase.

**3.** L37 … favour emission of potent greenhouse gases (GHG) e.g. $CH_4$ and $N_2O$

**Author's Response:** We have rewritten the introduction to improve its clarity. This paragraph was improved as follows:

L28-33. "Coastal wetlands are found at the interface of terrestrial and marine ecosystems and account for 10% of the global wetland area (Lehner and Döll 2004). They are highly productive and provide various ecosystem services such as water quality improvement, biodiversity, and carbon sequestration (Duarte et al, 2013). For instance, mangroves can accumulate five times more soil carbon than terrestrial forests (Kauffman et al, 2020). However, the high productivity and anoxic soil conditions that promote carbon sequestration can also favour potent greenhouse gas emissions (GHGs), including $CO_2$, $CH_4$ and $N_2O$ (Whalen, 2005; Conrad, 2009).

**4.** L44 Reference needed.

**Author's Response:** Reference was added.

L36-42. The emission of $CO_2$ is a result of respiration, where fixed carbon by photosynthesis is partially released back into the atmosphere (Oertel et al., 2016).

**5.** L48 convoluted sentence, please improve sentence structure

**Author's Response:** The sentence was improved as follows:

L43-45. "Thus, emissions of GHG from land-use change can be mitigated through the reversal of these activities, for instance, reduction of fertiliser use and the reinstallation of tidal flow on unused agricultural land (Rashti et al, 2016; Kroeger et al. 2017).."

**6.** L51-53 References needed

**Author's Response:** Reference Knox et al. (2015) was added to L45.

**7.** L60-61. References needed

**8.** The sentence was removed from the revised introduction.

**9.** L66-77 sentence does not flow well from the previous statement.

L66-67 Can you please make this nuanced to reflect that it is the balance between process rates and the area over which they occur determines the important for tropical regions net emissions.

**Author's Response:** We have rewritten the introduction to improve the flow of information and thesection about GHG emissions and their driving factors was fully revised as following:

L35-42. The GHG emissions in coastal wetlands primarily result from microbial processes in the soil-water-atmosphere interface (Bauza et al, 2002; Whalen, 2005). The emission of $CO_2$ is a result of respiration, where fixed carbon by photosynthesis is partially released back into the atmosphere (Oertel et al., 2016). Emissions of $CH_4$ result from anaerobic and aerobic respiration by methanogenic bacteria, mostly in waterlogged conditions (Angle et al., 2017; Saunois et al, 2020). Finally, $N_2O$ emissions are caused by denitrification in anoxic conditions and nitrification in aerobic soils, both driven by nitrogen content and soil moisture (Ussiri and Lal 2013). Thus, the total GHG emissions from a wetland are driven by environmental conditions that favour these microbial processes, all of which result in highly variable emissions from wetlands worldwide (Kirschke et al., 2013; Oertel et al. 2016).

**10.** L83-85 This sentence is not clear to me. Can you please improve the flow of the text.

**Author's Response**. We clarified the sentence in the revised introduction as described below:

L55-60. This study measured the annual GHG fluxes ($CO_2$, $CH_4$ and $N_2O$) from three natural coastal wetlands (mangroves, saltmarsh and freshwater tidal forests) and two agricultural land use sites (sugarcane plantation and pasture) in tropical Australia. The objectives were to assess the difference in GHG fluxes throughout different seasons that characterise tropical climates (dry-cool, dry-hot and wet-hot) and to identify environmental factors associated with these GHG fluxes. These data will inform emission factors for converting wetlands to agricultural land uses and vice versa, filling in a knowledge gap identified in Australia (Baldock et al., 2012) and tropical regions worldwide (IPCC, 2013).

**11.** You need to explain the rationale for high emissions during high tides. In the intro you agree that more sulphate reduce $CH_4$ production. These points seem contradictory to me. L131. Four or three sampling events? This is a bit unclear to me. Is it correct that you measured during different tides only once? You need to consider if that is enough in the context of seasonally. The tidal impacts are a bit unclear to me, from the final sentence in the introduction it sounds to me that all of the wetlands are impacted by tides? Please clarify this.

**Author's Response:** The measurements of low vs high tide was just a one-time additional experiment to verify that tide was not strongly affecting our sampling design We have clarified this in the method section and deleted this statement from the main hypothesis in the introduction.

L122-130. Additionally, we assessed the variability of our measurements with tidal inundation in mangroves and saltmarsh, which were regularly inundated (~10-30 cm). For this, we measured GHG emissions during a low (0.7m) and a high tide (2.8m; Lucinda, 18° 31' S; 146° 23 'E) in the dry-cool period of 2019. We found that $CH_4$ fluxes did not significantly vary between the low and high tide within all coastal wetlands. Contrarily, for saltmarsh, $CO_2$ was taken during the high tide (1.12 ± 0.24 g $m^{-2}$ $d^{-1}$) but emitted (0.69 ± 0.4 g $m^{-2}$ $d^{-1}$) during the low tide ($F_{1,28} = 20.06$, $p < 0.001$). Finally, for $N_2O$, fluxes differed in all coastal wetlands, with higher uptakes in the high tide for mangroves ($F_{1,28} = 38.28$, $p < 0.001$; $F_{1,28} = 13.53$, $p = 0.001$) and higher release for saltmarsh ($F_{1,28} = 38.31$, $p < 0.001$) during low tide (Table S4). These results suggested that for $CO_2$ and $N_2O$ fluxes, there was a probability of variation depending on the time of sampling. Thus, further sampling was conducted only during low tides.

**12.** I suggest you swap the order of section 2.2 and 2.3 as you refer to the gas chromatography set up in the current section 2.2

**Author's Response:** Thanks for the suggestion but we think that the current order goes well with the flow of information. Section 2.2 refers to a gas isotope ratio mass spectrometer (L-114) not a gas chromatograph (L151).

**13.** Section 2.3.

You need to include some detail on the spatial distribution of your samples. What is the size of the sampled area, and how did you determine if it is representative of other systems with similar land use? I have the feeling that there is a risk of pseudo replication but cannot assess that without some more detail. If you have subsamples within the same area rather than independent replicate samples from each land use class that need to be reflected in your conclusions. If you do not have independent replicates, you do not have the statistical basis for making statements relating to land use, you can only state that the sites are different so you need to be much more cautious in your recommendations in the discussion.

What method was used for the randomisation?

**Author's Response:**

We acknowledge the limitation of this study in terms of land use replication. For this study, we wanted to focus on addressing the small-scale variability of each land use and temporal variations. Furthermore, land-use level replication of our studies was limited due to inaccessibility of these sites due to permission for access into farms, adverse weather during most of the year (e.g. during very hot conditions or during flooding), safety risk due to crocodiles and the high cost of sample analysis (>AUD 8,000 per experiment). We included details about statistical analyses and replicates in supplementary file. As suggested, we toned down our recommendations including the title in the revised manuscript and throughout the discussion section as follows:

-300. This study supports the application of three management actions that could reduce GHG emissions. First, the conversion of ponded pastures to coastal wetlands is likely to reduce soil GHG emissions.

L30-308: Legal enablers in Queensland are in place to manage unproductive agricultural land this way (Bell-James and Lovelock 2019), and could provide an alternative income source for farmers.

L317-320. Future studies should increase the number of sites of ponded pastures to account for variability in hydrology, fertilisation, and cattle use. However, the very high difference (2-3 orders of magnitude) between dry and ponded pastures provides confidence that pasture management could provide significant GHG mitigation throughout the year.

**14.** How did you deal with areas with vegetation?

**Author's Response:**

We did not place incubation chambers on vegetation where possible because our objective was to measure GHG emissions from the soil. This was elaborated in the methods section as follows:

L136-138. Five chambers were set ~ 5cm deep in the soil at random locations one day before sampling to minimise the disturbance of installation during the experiment (Rashti et al, 2015).

The chambers were selectively located on soil with minimal vegetation, roots, and crab burrows.

**15.** What number of gas samples were collected from each chamber after the initial tests? During with season, did you test for linearity?

**Author's Response:**

Four samples were collected from each chamber at 0, 20, 40 and 60 minutes to measure the linearity of the GHG fluxes over time. However, for GHG flux calculations, we collected two samples from all five chambers at 0 and 60 minutes. Linearity was tested for all chambers during dry-hot seasons and one chamber per site for all other seasons. Our previous experience with this method has taught us that this is the most cost-effective way to measure GHG from wetlands (Kavehei et al., 2021) and agricultural lands (Rashti et al., 2015). Linearity results were provided in supplementary files (S2). We clarified this in the manuscript as follows:

L142-146. At the start of the experiment, gas chambers were closed. A sample was taken at time zero and then after one hour with a 20 ml syringe and transferred to a 12 mL-vacuumed exetainer (Exetainer, Labco Ltd., High Wycombe, UK). During the dry-hot season, linearity tests of GHG fluxes with time were conducted by sampling at 0, 20, 40 and 60 min (Rashti et al, 2016). For the rest of the experiments, linearity tests were performed in one of the five chambers at each site; $R^2$ values were consistently above 0.70.

**16.** I think you may well have impacts of ebullition of CH4, there is signs of that in Figure 2. If you could not test for linearity for $CH_4$ fluxes especially during the flooded period, your fluxes may not be correct.

**Author's Response:**

Yes. Methane ebullition effects were reflected in the wet pasture ecosystem through high emissions. We measured the linearity of one chamber in each site for three days in the flooded period to present precise fluxes, and the $R^2$ value was ranged between 0.6-0.9 (except for CO2 in marsh and N2O in mangroves, SEE S2). This was described in the manuscript as following:

L145-146. For the rest of the experiments, linearity tests were performed in one of the five chambers at each site $R^2$ values were mostly above 0.70.

**17.** Some of your areas looks as if they have standing water, how did you sample gas fluxes on these? Did you use floating chambers? Please add more detail about the sampling.

**Author's Response:** Our sites in mangroves, saltmarsh and ponded pasture ecosystem had always standing water; however, the water was never deep enough to require floating chambers. Therefore, we used the static chambers but with the lateral holes opened to allow water movement and with vertical extension to avoid full submersion. We have added the details in the manuscript as follows:

L132-136. "We used static, manual gas chambers made of high-density, round polyvinyl chloride pipe, which consisted of two units: a base (r =12 cm, h =18 cm) and a detachable collar (h =12 cm; Hutchinson and Mosier, 1981; Kavehei et al, 2021). The chambers had lateral holes that could be left covered with rubber bungs at low water levels and left open at high water levels to allow for water movement between sampling events. When the wetlands were inundated for the experiments, we used PVC extensions (h = 18 cm)."

**18.** Your statistics are not clear to me. Please add some more detail to make it clear how you analysed for variation and interactions between the two main factors in your study site and season.

**Author's Response:**

Variation and interactions between the two main factors, e.g., site and season, were analysed through Kruskal-Wallis test and Mann-Whitney U Test when data did not comply with the assumptions of normality. One-way Analyses of Variance (ANOVA) was used for normally distributed data to analyse the difference between sites and seasons. Details were added in the manuscript as follows:

L167-174. When data were not normal, they were transformed (log, 1/x) to comply with the assumptions of normality and homogeneity of variances. Despite transformations, some variables were not normally distributed; thus, the differences between sites and seasons were analysed with the non-parametric Kruskal-Wallis test and Mann-Whitney U Test. The data which met the normality assumptions were analysed for spatial and temporal differences with

one-way Analyses of Variance (ANOVA), where site and season were the predictive factors and replicate (gas chamber) was the random factor of the model. Additionally, a Pearson correlation test was run to evaluate the correlation of GHG with measured environmental factors.

**19.** L185. what is the assumption of w=17.38 for each season based on? This needs to be justified in the context of seasonal climate data.

**Author's Response:**

On the basis of 40 years of climate data on mean maximum temperature and rainfall from the Bureau of metrology Australia (Bureau of Meteorology, http://www.bom.gov.au/jsp/ncc/cdio/cvg/av), we assumed that each season consisted of ~17.38 weeks. The source file was attached as a supplementary file (S3).

**20.** What are the number of temporal replicates within each season n=1?

**Author's Response:** Within each season, we measured each site for at least three days. This was described in methods section as following:

L119-120. We measured GHG fluxes ($CO_2$, $CH_4$ and $N_2O$) at each site for three consecutive days during each sampling period except for the dry-cool period of 2018, when mangroves, saltmarsh and sugarcane were surveyed for one day.

**21.** Is table 3 the same data as in Figure 2. If so, I suggest not showing the same data twice. If different, please make captions and table headings clearer to help the reader understand the data.

**Author's Response:**

Thanks for the suggestion. Table 3 was excluded from the revised manuscript and added in supplementary file as Table 2S.

**22.** L281-282-286 Here you are repeating results in the discussion. I suggest you focus this part of the text to compare and contrast to other studies.

**Author's Response:**

This section was modified to provide the trends to compare and contrast with literature. The manuscript was modified as follows.

L191-197. Soil emissions for $CO_2$ were significantly different among sites and times of the year ($t$ =155.09, n =237, $p < 0.001$; Fig. 2a). The highest $CO_2$ emissions were measured during the wet-hot period in the dry pasture, where values reached $20.31 \pm 1.95$ g m$^{-2}$ d$^{-1}$ while the lowest values were measured in the saltmarsh, the only site that acted as a sink of $CO_2$ with an uptake rate of $-0.59 \pm 0.15$ g m$^{-2}$ d$^{-1}$. In the pastures, $CO_2$ emissions were twice as high when dry with cumulative annual emissions of $5,748 \pm 303$ g m$^{-2}$ y$^{-1}$ compared to when wet, with $2,163 \pm 465$ g m$^{-2}$ y$^{-1}$. For the coastal wetlands, cumulative annual $CO_2$ emissions were highest in freshwater tidal forests with $2,213 \pm 284$ g m$^{-2}$ y$^{-1}$, followed by mangroves with $1,493 \pm 111$ g m$^{-2}$ y$^{-1}$ and lowest at the saltmarsh with uptake rates of $-264 \pm 29$ g m$^{-2}$ y$^{-1}$.

**23.** L290 You state here that temperature is a driver of the fluxes you measured but your stats does not support this, i.e. no significant effect so I don't think this point is valid in the light of your results.

**Author's Response:** The point was clarified in the revised paragraph as follows:

The variability of GHG fluxes was best explained by land use and wetland type; however, some trends with seasons were evident. For instance, $CO_2$ and $N_2O$ emissions were lowest during the dry-cool periods. L258-260.

**24.** L298 All your chambers were dark – I do not get the point of this statement. Why single out mangroves.

**Author's Response:**

Thanks for highlighting this point. The statement was excluded from the revised manuscript.

**25.** L300. High $CH_4$ emissions during the hot-dry season – How dry were the soil? Or were they sit wet in the high emitting sites?

**Author's Response:**

All measured sites in coastal wetlands and wet pasture were always wet, even during the hot-dry season. High $CH_4$ emissions during the hot-dry season were discussed as follows;

L249-251. Emissions of $CH_4$ also tend to increase with temperature as the activity of soil methane-producing microbes (Ding et al, 2004) and the availability of carbon is higher in warmer conditions (Yvon-Durocher et al, 2011).

**26.** I think the paper need to include some data in the environmental conditions measured in the different seasons to understand what conditions the microorganisms were experiencing.

**Author's Response:** We included the detailed information on the environmental factors and GHG emissions as a supplementary file because none of the measured main influencing factors (including temperature, rainfall, water-filled pore space and bulk density) was correlated with GHG emissions. We mentioned in the manuscript as following:

L220-225. Overall, we found that not one single parameter measured in this could explain GHG fluxes for all sites except land-use. The $CO_2$ emissions were not significantly correlated to bulk density ($R^2 = 0.026$ $p = 0.918$ $n =18$), % WFPS ($R^2 =$-0.003 $p = 0.99$ $n = 18$), or soil temperature ($R^2 = 0.296$ $p = 0.233$, $n =18$). Soil $CH_4$ emissions were neither correlated with bulk density ($R^2 = $-0.096 $p = 0.706$ $n = 18$), % WFPS ($R^2 = 0$-.224 $p = 0.372$, $n = 18$) or soil temperature ($R^2 = 0.286$ $p = 0.25$ $n = 18$). Finally, no correlation was found between $N_2O$ emissions and bulk density ($R^2 = $-0.349 $p = 0.156$ $n =18$), % WFPS ($R^2 = $-0.34 $p = 0.168$ $n =18$), or soil temperature ($R^2 = $-0.241 $p = 0.335$ $n = 18$; S4). See full raw dataset at Table 1S and S4..

**27.** Management implications section

Since I do not think you have independent replication (at least I cannot determine if you do from the methods section) makes it hard to make strong conclusions about land use. As I mentioned earlier you can only state that you have differences between sites but not link these differences specifically to land use as other site specific effect may cause these differences.

**Author's Response:** Following your suggestions, the conclusions were modulated by modifying the discussion section as following:

L330-336. The GHG emissions from three coastal wetlands in tropical Australia (mangroves, saltmarsh and freshwater tidal forests) were consistently lower than those from two common agricultural land use of the region (sugarcane and pastures) throughout three climatic conditions (dry-cool, dry-hot and wet-hot). Ponded pastures, which emitted 200 times more $CH_4$, and sugarcane emitted seven times more than any natural coastal wetland. If these high emissions are persistent in other locations and within other tropical regions, conversion of pastures and sugarcane to similar coastal wetlands could provide significant GHG mitigation.

As nations try to reach their emission reduction targets, projects aimed at converting or restoring coastal wetland can financially benefit farmers and provide additional co-benefits derived from coastal wetland restoration.

**28.** In the discussion, I think it is important to consider if your space for time model is valid, i.e. is it plausible that the current agricultural system would revert to function as the natural system you measured fluxes from? This needs careful discussion as ecosystem restoration does often not take you back to the starting point, or at least it can take a long time for the restored system to regain its original functions.

**Author's Response:**

The potential for GHG mitigation for changing agricultural lands to wetlands is promising; however, there is still uncertainty of whether degraded land can be successfully reverted to wetlands. It is likely that, instead, a new type of ecosystem could be created (Hobbs et al. 2009), and that legacy of land use could last for years (Ardon et al. 2017). However, this study suggests that this potential should be further explored in similar land uses in tropical regions. Additionally, future monitoring of newly created wetlands would provide information on whether and when the full GHG mitigation can be achieved through wetland creation or restoration.

**29.** L304-309 You have not measured these parameters so you can only speculate that they cause low emissions. The way this statement is phrased suggests your study has demonstrated this which is not the case. Please rephrase.

**Author's Response**: The paragraph was rephrased as follows:

L257-266. Notably, coastal wetlands, even the freshwater tidal forests, had much lower emissions compared to the wet pastures. This large difference could be attributed to the presence of terminal electron acceptors in the soils (e.g. iron, sulphate, manganese) of the coastal wetlands, which could inhibit methanogenesis (Kögel-Knabner et al, 2010; Sahrawat, 2004). Sulphate reducing bacteria are also likely to outcompete methane-producing bacteria (methanogens) in the presence of high sulphate concentrations in tidal wetlands, resulting in low $CH_4$ production. Competition between methanogens and methanotrophs may result in a net balance of low $CH_4$ production despite freshwater conditions (Maietta et al. 2020). Additionally, microorganism living within the bark of *Melaleuca* trees can consume $CH_4$ (Jeffrey et al, 2021), so it is possible that similar bacteria within the soil could reduce $CH_4$

emissions. Interestingly, variability within $CH_4$ fluxes among sites was very high, despite them being very close to each other (Fig. 1b). These differences highlight the importance of land use in GHG fluxes, which are likely to significantly alter the microbial community composition and abundance, which can change rapidly over small spatial scales (Martiny et al, 2006; Drenovskyet al, 2009).

**30.** Describe how you calculate your $CO_2$eq in the methods section and present this in the results before discussing these data.

**Author's Response:**

We described the $CO_2$eq calculation method in the methods section as following:

L165-167. The $CO_{2\text{-equivalent}}$ ($CO_{2\text{-eq}}$) values were estimated by multiplying $CH_4$ and $N_2O$ emissions by 25 and 298, respectively (Solomon, 2007), which represent the radiative balance of these gases (Neubauer, 2021).

**31.** Plant mediated emissions of $CH_4$ and $N_2O$ are likely to be important in your system. As this would impact on your overall conclusion regarding the global warming potential of the different sites I think you need to discuss this.

**Author's Response:**

This is a good point. The following information was added to discuss the suggested point.

L284-289. However, the GHG flux measurements from this study did not account for the effects of vegetation, which can alter fluxes. For instance, some plant species of rice paddies (Timilsina et al., 2020) and *Miscanthus sinensis* (Lenhart et al., 2019) can increase $N_2O$ emissions, and some tree species can facilitate $CH_4$ efflux from the soil (Pangala et al. 2013). Finally, changes in emissions between low and high tides were detected for $CO_2$ and $N_2O$. Thus, future studies that include vegetation and changes within tidal cycles will improve GHG flux estimates for coastal wetlands.

**32.** After reading the manuscript, I think that although it presents novel data, I do think it is premature to use the manuscript in its current form as a basis for management recommendations.

**Author's Response:**

We appreciate your feedback to improve the present version of the manuscript. We tried our best to incorporate the suggested revisions. We have highlighted in the manuscript the limitations of our sampling and modified the management recommendations and conclusions accordingly.

---

## Author Response (AR2)

I am pleased with the author's response to my earlier comments. I think the manuscript is significantly improved. However, I still have a few specific suggestions below for authors, so the paper will be impactful. Once the authors have addressed these comments, I think the manuscript is about ready to be accepted.

**Author's response**: We thank anonymous referee#1 for constructive feedback and for highlighting the improvement in the quality of the revised manuscript. The received recommendations were carefully considered and incorporated into the current version of the manuscript. A point-by-point response to comments was given below. Please note that line numbers in the author's response refer to line numbers in the revised manuscript. And changes made in the original manuscript were highlighted in yellow.

**1.** Please consider double-checking and improve all intext referencing following journal style. I found some of them are not consistent.

**Author's response**: We have carefully checked all references and aligned them with journal-style with consistency.

**2.** I think that authors need to add a few sentences highlighting the limitations of this study somewhere in the methods or discussions, particularly regarding sampling replicates. I know that the authors have addressed my previous comment regarding time, safety and funding limitations to perform robust spatial and temporal GHG measurement replicates. Describing limitations in the methods section will not downgrade your paper; as an ecologist, we know how hard to work in wetland ecosystems.

**Author's response**: We have described the study limitations in the methods section:
Lines 159-163: For annual cumulative soil GHG flux calculations from coastal wetlands, we used GHG fluxes measured during low tide; therefore, our values did not incorporate the effect of tidal fluctuations. The spatial and temporal replication of this study targeted spatial variation within soil type ($< 50$ cm, five chambers), days (three days per sampling) and seasons (three seasons per year). However, our replication within land use and wetland type was limited; thus, generalisations for all wetlands and land uses should be done acknowledging this limitation.

**3.** Lines 159: Please add a unit for "Seasonal cumulative GHG fluxes" in Equation 1. For this equation, I also wonder if the authors could provide clarification regarding the incorporation of the tidal factor for cumulative GHG calculations. Please note that sampling was done during low tides only as described in line 131, particularly for CO2 and N2O in samples coastal wetlands.

**Author's response**: The suggested units were added in Equation 2 and clarification about GHG flux measurement in low tide were made as following: L159-154.

$$\text{Seasonal cumulative GHG fluxes (mg or } \mu g\ m-2\ yr-1) = \sum_{i=1}^{n} (R_i \times 24 \times D_i \times 17.38)$$

Equation 2

Where;

$R_i$ = Gas emission rate (mg m$^{-2}$ hr$^{-1}$ for $CO_2$ and μg m$^{-2}$ hr$^{-1}$ for $CH_4$ and $N_2O$) during low tide

$D_i$ = Mean daily GHG emission rate in a season, (mg m$^{-2}$ d$^{-1}$ for $CO_2$ and μg m$^{-2}$ d$^{-1}$ for $CH_4$ and $N_2O$)

17.38 = number of weeks in each season, assuming these conditions were representative of the annual cycle (see Table 1). 2

**4.** Section 3.3: put comma after R2, p-value, and n.
**Author's Response**: Section 3.3 was revised according to the recommendation.

**5.** Line 279-290: This is a really good paragraph to compare the authors' findings with the measurements elsewhere. However, it may be hard to understand for non-technical readers. I suggest authors add publishes numbers from a quick literature survey and 2013 IPPC wetlands supplement and compile them (including GHG numbers in this study) in the summary table. This Table will be useful to connect these study findings with the next policy implication discussion in Section 4.1. The Table should present GHG data along with their 95% CI (common uncertainty used by IPCC) and sample size (n).

**Author's response**: The following literature review table was added as suggested and text in the paragraph was modified accordingly as following:

L281-284 and L295-298: The fluxes measured in the coastal wetlands of this study, -1191 to 10,970 mg m$^{-2}$ d$^{-1}$ for $CO_2$, -0.3 to 3.9 mg m$^{-2}$ d$^{-1}$ for $CH_4$, and -0.2 to 2.8 mg m$^{-2}$ d$^{-1}$ for $N_2O$, were within the range of those measured in other subtropical/tropical wetlands, worldwide (Table 3). For $CO_2$, fluxes can range between 44 and 11,328 mg m$^{-2}$ d$^{-1}$, for $CH_4$, from 0.03 to 1255 mg m$^{-2}$ d$^{-1}$ and for $N_2O$, from 0.1 to 279 mg m$^{-2}$ d$^{-1}$ (Table 3). Despite being in tropical regions, GHG fluxes from this study were lower compared to other climates (Table 3).

Table: 3. Comparison of GHG fluxes (mg m$^{-2}$ d$^{-1}$) from literature review with the rates reported in this study

| Reference | Climate | Country | Ecosystem | $CO_2$ fluxes | $CH_4$ fluxes | $N_2O$ fluxes |
|---|---|---|---|---|---|---|
| Allen et al., 2011 | Subtropical | Australia | Mangroves estuary | - | 1.5-51 | - |
| Cabezas et al., 2018 | Subtropical | USA | Mangroves estuary | - | 0.3-2.2 | - |
| Li and Mitsch, 2016 | Subtropical | USA | Flooded brackish marsh | - | $212 \pm 51$ | - |
| Morse, Ardón and Bernhardt, 2012 | Subtropical | USA | Forested wetlands | 7224-11328 | 118-1255 | 46-279 |
| Musenze et al., 2014 | Subtropical | Australia | Mangroves estuary | - | 5-448 | 0.1-3.4 |
| Whiting and Chanton, 2001 | Subtropical | USA | Typha marsh | 409-477 | 189-264 | - |
| Mitsch et al., 201 | Tropical | South Africa | Seasonally flooded wetland | - | 264±29 | - |

| Reference | Climate | Country | Ecosystem | | | |
|---|---|---|---|---|---|---|
| Krithika et al., 2008 | Tropical | India | Mangroves | - | 25-50 | - |
| Kristensen et al., 2008 | Tropical | Tanzania | Mangroves | 44-3521 | 1.9-6.5 | - |
| Biswas et al., 2007 | Tropical | India | Mangroves estuary | - | 0.03-2.16 | - |
| Purvaja et al., 2004 | Tropical | India | Mangroves estuary | - | 10-85 | - |
| Kreuzwieser et al., 2003 | Tropical | Australia | Mangroves | - | 0.6-11 | - |
| Kiese and Butterbach-Bahl, 2002 | Tropical | Australia | Tropical rain forest | 2208-3288 | | 1.9-3.2 |
| Purvaja and Ramesh, 2000 | Tropical | India | Mangroves | - | 63-434 | - |
| Sotomayor et al. 1994 | Tropical | Puerto Rica | Mangroves | - | 5-110 | - |
| Barnes et al., 2006 | Tropical | India | Mangroves | - | 9-15 | - |
| Melling et al. 2012 | Tropical | Malaysia | Peat swamp forest | 3384 | 21-29 | |
| This study | Tropical | Australia | Freshwater tidal forest | 1640-10970 | 0.16-0.59 | -0.19-0.7 |
| This study | Tropical | Australia | Saltmarsh | -594-(-1191) | -0.25-0.12 | -0.22-2.76 |
| This study | Tropical | Australia | Mangroves | 2852-5669 | 0.44-3.95 | 0.04-0.16 |

| Oertel et al., 2016 | (Sub) Tropical | Global | Wetlands | - | -1.08-1169 | - |
|---|---|---|---|---|---|---|
| Oertel et al., 2016 | Temperate | Global | Wetlands | - | -1.49-1510 | - |
| Oertel et al., 2016 | Mediterranean | Global | Wetlands | - | - | -2.6-9.4 |
| Al-Haj and Fulweiler, 2020 | - | Global | Mangroves | - | -1.1-1169 | -0.2-6.3 |
| Al-Haj and Fulweiler, 2020 | - | Global | Saltmarshes | 6844-34983 | 0.38-3002 | -7.39-28.52 |
| Rosentreter et al., 2021 | - | Global | Mangroves | 4563-30800 | -0.69-10.78 | -1.69-4.65 |
| Rosentreter et al., 2021 | - | Global | Saltmarshes | 3802-20914 | 107-168 | 4.96 |
| IPCC 2013 | Tropical | Global | Swamp forests | | 30.76-2149 | |

Note: Hyphen means no data was available; GHG fluxes as $CO_2$-C, $CH_4$-C and $N_2O$-N were multiplied by 3.66, 1.34 and 1.57 respectively to calculate $CO_2$, $CH_4$ and $N_2O$ fluxes (National Greenhouse Accounts Factors, Australian Government Department of Industry, Science, Energy and Resources. 2020).

**6.** Supplement Table 1S: I suggest authors provide sampling dates for their raw data.

**Author's response**: We submitted the revised for their raw data with sampling dates includes.

**Responses to the Comments from refree#2**

Dear Authors. Thank you for the revised manuscript. I found that you have done a thorough job. I have a few further recommendations.

**Author's response**: We thank the Anonymous referee#2 for useful recommendations and

We have included a point-by-point response to comments raised by the reviewer, and a revised manuscript has been submitted.

1. At the moment you report some results on the GHGs in your methods section. I recommended that you change this so that the results are presented in the results section.

**Author's Response:** We moved the result reporting text to the results section as following:

L210-215. We found that $CH_4$ fluxes did not significantly vary between the low and high tide within all coastal wetlands. Contrarily, for saltmarsh, $CO_2$ was taken during the high tide (1.12 $\pm$ 0.24 g m$^{-2}$ d$^{-1}$) but emitted (0.69 $\pm$ 0.4 g m$^{-2}$ d$^{-1}$) during the low tide ($F_{1,28} = 20.06$, $p < 0.001$). Finally, for $N_2O$, fluxes differed in all coastal wetlands, with higher uptakes in the high tide for mangroves ($F_{1,28} = 38.28$, $p < 0.001$; $F_{1,28} = 13.53$, $p = 0.001$) and higher release for saltmarsh ($F_{1,28} = 38.31$, $p < 0.001$) during low tide (Table S4). These results suggested that for $CO_2$ and $N_2O$ fluxes, there was a probability of variation depending on the time of sampling.

2. You mention that you sample in five locations at each site and that your aim was to assess small scale variability. I could not find an evaluation of the small scale variability apart from SEs reported, so I am not sure how this is brought to the paper. I also could not find the details of the distance between the sampling points within each location. I think that this is important that needs to be included as it will help the readers to understand how the variability you found relates to the spatial distribution of your sampling points. I also think it would be valuable for the reader to understand how you selected your sampling points within each site, i.e. how did you do your randomisation of the selected points within site, taking into account your logistical constraints.

**Author's Response:** Please refer to the following text:

L77: The natural coastal wetlands and the sugarcane site were located within the same property at Insulator Creek and w**ere located < 200 m apart** (Fig. 1B), while the ponded pasture was 20 km north at Mungalla Station (Fig. 1A)

 L132: Five chambers were set ~ 5cm deep in the soil separated **one to two meters** from each other selectively located on soil with minimal vegetation, roots, and crab burrows.

    **3.** Also, I found a few places where the sentences where not quite right (e.g. no capital at the start of a sentence) so I recommend that you proof read the revised text to remove such small mistakes.

**Author's Response:** We proofread the manuscript to remove the mistakes.

---

## Author Response (AR3)

Dear Sara,

Thanks for raising an important question. We addressed the negative uptake by saltmarsh soil and revised the manuscript accordingly as following:

L281-289: The fluxes measured in the coastal wetlands of this study, -1191 to 10,970 mg m$^{-2}$ d$^{-1}$ for $CO_2$, -0.3 to 3.9 mg m$^{-2}$ d$^{-1}$ for $CH_4$, and -0.2 to 2.8 mg m$^{-2}$ d$^{-1}$ for $N_2O$, were within the range of those measured in other subtropical/tropical wetlands, worldwide (except for the negative $CO_2$ fluxes in saltmarsh soils, Table 3). For $CO_2$, fluxes can range between 44 and 11,328 mg m$^{-2}$ d$^{-1}$, for $CH_4$, from 0.03 to 1255 mg m$^{-2}$ d$^{-1}$ and for $N_2O$, from 0.1 to 279 mg m$^{-2}$ d$^{-1}$ (Table 3). Despite being in tropical regions, GHG fluxes from this study were lower compared to other climates (Table 3). Contrary to previous studies, $CO_2$ uptake by saltmarsh soil was likely to be linked with dark $CO_2$ fixation in wetland soils (Akinyede et al., 2020; Mar Lynn et al., 2017). Wetland soils exhibit autotrophic bacteria which contribute to dark $CO_2$ fixation at ~311 mg m$^{-2}$ d$^{-1}$ however these rates could vary depending upon abundance and diversity of microbial communities ((Akinyede et al., 2020). Further studies exploring presence and abundance of $CO_2$ fixing bacteria in saltmarsh soils is recommended.